# Supervised Training of Conditional Monge Maps

**Charlotte Bunne**[*]
ETH Zurich
bunnec@ethz.ch

**Andreas Krause**
ETH Zurich
krausea@ethz.ch

**Marco Cuturi**
Apple
cuturi@apple.com

## Abstract

Optimal transport (OT) theory describes general principles to define and select, among many possible choices, the most efficient way to map a probability measure onto another. That theory has been mostly used to estimate, given a pair of source and target probability measures $(\mu, \nu)$, a parameterized map $T_\theta$ that can efficiently map $\mu$ onto $\nu$. In many applications, such as predicting cell responses to treatments, pairs of input/output data measures $(\mu, \nu)$ that define optimal transport problems do not arise in isolation but are associated with a *context c*, as for instance a treatment when comparing populations of untreated and treated cells. To account for that context in OT estimation, we introduce CONDOT, a multi-task approach to estimate a family of OT maps conditioned on a context variable, using several pairs of measures $(\mu_i, \nu_i)$ tagged with a context label $c_i$. CONDOT learns a *global* map $\mathcal{T}_\theta$ conditioned on context that is not only expected to fit *all labeled pairs* in the dataset $\{(c_i, (\mu_i, \nu_i))\}$, i.e., $\mathcal{T}_\theta(c_i)\sharp\mu_i \approx \nu_i$, but should also *generalize* to produce meaningful maps $\mathcal{T}_\theta(c_{\text{new}})$ when conditioned on unseen contexts $c_{\text{new}}$. Our approach harnesses and provides a novel usage for *partially input convex neural networks*, for which we introduce a robust and efficient initialization strategy inspired by Gaussian approximations. We demonstrate the ability of CONDOT to infer the effect of an arbitrary combination of genetic or therapeutic perturbations on single cells, using only observations of the effects of said perturbations separately.

## 1 Introduction

A key challenge in the treatment of cancer is to predict the effect of drugs, or a combination thereof, on cells of a particular patient. To achieve that goal, single-cell sequencing can now provide measurements for individual cells, in treated and untreated conditions, but these are, however, not in correspondence. Given such examples of untreated and treated cells under different drugs, can we predict the effect of new drug combinations? We develop a general approach motivated by this and related problems, through the lens of *optimal transport (OT) theory*, and, in that process, develop tools that might be of interest for other application domains of OT. Given a collection of $N$ pairs of measures $(\mu_i, \nu_i)$ over $\mathbb{R}^d$ (cell measurements), tagged with a context $c_i$ (encoding the treatment), we seek to learn a context-dependent, parameterized transport map $\mathcal{T}_\theta$ such that, on training data, that map $\mathcal{T}_\theta(c_i) : \mathbb{R}^d \to \mathbb{R}^d$ fits the dataset, in the sense that $\mathcal{T}_\theta(c_i)\sharp\mu_i \approx \nu_i$. Additionally, we expect that this parameterized map can generalize to unseen contexts and patients, to predict, given a patient's cells described in $\mu_{\text{new}}$, the effect of applying context $c_{\text{new}}$ on these cells as $\mathcal{T}_\theta(c_{\text{new}})\sharp\mu$.

**Learning Mappings Between Measures** From generative adversarial networks, to normalizing flows and diffusion models, the problem of learning maps that move points from a source to a target distribution is central to machine learning. OT theory (Santambrogio, 2015) has emerged as a principled approach to carry out that task: For a pair of measures $\mu, \nu$ supported on $\mathbb{R}^d$, OT suggests that, among all maps $T$ such that $\nu$ can be reconstructed by applying $T$ to every point in the support

---

[*]Work done during an internship at Apple.

**a.** ... by scalar

e.g., time-course or dosage levels

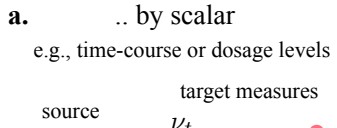

.. learn $\mathcal{T}_\theta(t)_\sharp\mu$

**b.** ... by covariate

e.g., metadata or identifiers

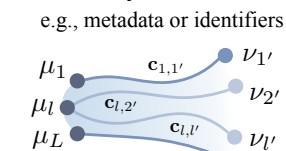

.. learn $\mathcal{T}_\theta(\mathbf{c}_{l,l'})_\sharp\mu$

**c.** ... by action

e.g., perturbations or decisions

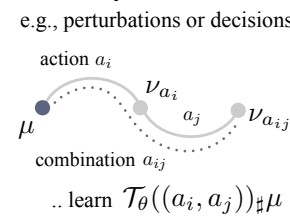

.. learn $\mathcal{T}_\theta((a_i, a_j))_\sharp\mu$

Figure 1: The evolution from a source $\mu$ to a target measure $\nu$ can depend on context variables $c$ of various nature. This comprises **a.** scalars such as time or dosage $t$ which determine the magnitude of an optimal transport, **b.** flow of measures into another one based on additional information (possibly different between $\mu$ and $\nu$) stored in vectors $\mathbf{c}_{l,l'}$, or **c.** discrete and complex actions $a_i$, possibly in combination $a_{ij}$. We seek a unified framework to produce a map $\mathcal{T}_\theta(c)$ from any type of condition $c$.

of $\mu$ (abbreviated with the push-forward notation as $T\sharp\mu = \nu$), one should favor so-called Monge maps, which *minimize* the average squared-lengths of displacements $\|x - T(x)\|^2$. A rich literature, covered in Peyré and Cuturi (2019), addresses computational challenges of estimating such maps, with impactful applications to various areas of science (cf., Hashimoto et al., 2016; Schmitz et al., 2018; Schiebinger et al., 2019; Yang et al., 2020; Janati et al., 2020; Bunne et al., 2022a).

**Neural OT** We focus in this work on neural approaches that parameterize the optimal maps $T$ as neural networks. An early approach is the work on Wasserstein GANs (Arjovsky et al., 2017), albeit the transport map is not explicitly estimated. Several recent results have exploited a more explicit connection between OT and NNs, derived from the celebrated Brenier theorem (1987), which states that Monge maps are necessarily gradients of convex functions. Such convex functions can be represented using input convex neural networks (ICNN) (Amos et al., 2017), to parameterize either the Monge map (Jacob et al., 2018; Yang and Uhler, 2019; Bunne et al., 2021, 2022b) or a dual potential (Makkuva et al., 2020; Korotin et al., 2020) as, respectively, the gradient of an ICNN or an ICNN itself. In this paper, we build on this line of work, but substantially generalize it, to learn a *parametric* family of context-aware transport maps, using a collection of labeled pairs of measures.

**Contributions** We propose a framework that can leverage *labeled* pairs of measures $\{(c_i, (\mu_i, \nu_i))\}_i$ to infer a *global* parameterized map $\mathcal{T}_\theta$. Hereby, the context $c_i$ belongs to an arbitrary set $\mathcal{C}$. We construct $\mathcal{T}_\theta$ so that it should be able, given a possibly unseen context label $c \in \mathcal{C}$, to output a map $\mathcal{T}_\theta(c) : \mathbb{R}^d \to \mathbb{R}^d$, that is itself the gradient of a convex function. To that end, we propose to learn these parameterized Monge maps $\mathcal{T}_\theta$ as the gradients of partially input convex neural networks (PICNN), which we borrow from the foundational work of Amos et al. (2017). Our framework can be also interpreted as a hypernetwork (Ha et al., 2016): The PICNN architecture can be seen as an ICNN whose weights and biases are *modulated* by the context vector $c$, which parameterizes a *family* of convex potentials in $\mathbb{R}^d$. Because both ICNN —and to a greater extent PICNN— are notoriously difficult to train (Richter-Powell et al., 2021; Korotin et al., 2020, 2021), we use closed-form solutions between Gaussian approximations to derive relevant parameter initializations for (P)ICNNs: These choices ensure that, *upon initialization*, the gradient of the (P)ICNNs mimics the affine Monge map obtained in closed form between Gaussian approximations of measures $\mu_i, \nu_i$ (Gelbrich, 1990). Our framework is applied to three scenarios: Parameterization of transport through a real variable (time or drug dosage), through an auxiliary informative variable (cell covariates) and through action variables (genetic perturbations in combination) (see Fig. 1). Our results demonstrate the ability of our architectures to better capture on out-of-sample observations the effects of these variables in various settings, even when considering never-seen, composite context labels. These results suggest potential applications of conditional OT to model personalized medicine outcomes, or to guide novel experiments, where OT could serve as a predictor for never tested context labels.

## 2 Background on Neural Solvers for the 2-Wasserstein Problem

**Optimal Transport** The Monge problem between two measures $\mu, \nu \in \mathcal{P}(\mathbb{R}^d)$, here restricted to measures supported on $\mathbb{R}^d$ and compared with the squared Euclidean metric, reads

$$T^\star := \arg\inf_{T\sharp\mu=\nu} \int_{\mathbb{R}^d} \|x - T(x)\|^2 d\mu(x). \tag{1}$$

The existence of $T^\star$ is guaranteed under fairly general conditions (Santambrogio, 2015, Theorem 1.22), which require that $\mu$ and $\nu$ have finite $L_2$ norm, and that $\mu$ puts no mass on $(d-1)$ surfaces of class $\mathcal{C}_2$. This can be proved with the celebrated Brenier theorem (1987), which states that there must exist a unique (up to the addition of a constant) potential $f^\star : \mathbb{R}^d \to \mathbb{R}$ such that $T^\star = \nabla f^\star$. This theorem has far-reaching implications: It is sufficient, when seeking optimal transport maps, to restrict the computational effort to seek a "good" convex potential, such that its gradient pushes $\mu$ towards $\nu$. This result has been exploited to propose OT solvers that rely on input convex neural networks (ICNNs) (Amos et al., 2017), introduced below

$$f^\star := \arg \sup_{f \text{ convex}} \mathcal{E}_{\mu,\nu}(f) := \int_{\mathbb{R}^d} f^* \mathrm{d}\mu + \int_{\mathbb{R}^d} f \mathrm{d}\nu. \tag{2}$$

In practice, Monge maps can be estimated using a dual formulation (Makkuva et al., 2020; Korotin et al., 2020; Bunne et al., 2022b; Alvarez-Melis et al., 2021; Mokrov et al., 2021). Indeed, $T^\star$ in (1) is recovered as $\nabla f^\star$, where $f^\star$ is defined in (2), writing $f^*$ for the Legendre transform of $f$.

**Convex Neural Architectures** Input convex neural networks (ICNN) are neural networks $\psi_\theta$ that admit certain constraints on their architecture and parameters $\theta$, such that their output $\psi_\theta(x)$ is a convex function of their input $x$ (Amos et al., 2017). As a result, they have been increasingly used as drop-in replacements to the set of admissible functions in (2). Practically speaking, an ICNN is a $K$-layer, fully connected network such that, at each layer index $k$ from 0 to $K-1$, a hidden state vector $z_k$ is defined recursively as in (3),

$$z_{k+1} = \sigma_k(W_k^x x + W_k^z z_k + b_k) \tag{3}$$

and $\psi_\theta(x) = z_K$, where, by convention, $z_0$ and $W_0^z$ are 0; $\sigma_k$ are *convex* non-decreasing activation functions; $\theta = \{b_k, W_k^z, W_k^x\}_{k=0}^{K-1}$ are the weights and biases of the neural network. While ample flexibility is provided to choose dimensions for intermediate hidden states $z_k$, the last layer must necessarily produce a scalar, hence $W_{K-1}^x$ and $W_{k-1}^z$ are line vectors and $b_{K-1} \in \mathbb{R}$. ICNNs are characterized by the fact that all weight matrices $W_k^z$ associated to latent representations $z$ must have *non-negative* entries. This, along with the specific activation functions, ensures the convexity of $\psi_\theta$. We encode this constraint by identifying these matrices as the elementwise softplus or ReLU of other matrices of the same size, or, alternatively, using a regularizer that penalizes the negative entries of these matrices. Since the work by Amos et al. (2017), convex neural architectures have been used within the context of OT to model convex dual functions (Makkuva et al., 2020), or normalizing flows derived from convex potentials (Huang et al., 2021). Their expressivity and universal approximation properties have been studied by Chen et al. (2019), who show that any convex function over a compact convex domain can be approximated in sup norm by an ICNN.

## 3 Supervised Training of Conditional Monge Maps

We are given a dataset of $N$ pairs of measures, each endowed with a label, $(c_i, (\mu_i, \nu_i)) \in \mathcal{C} \times \mathcal{P}(\mathbb{R}^d)^2$. Our framework builds upon two pillars: (i.) we formulate the hypothesis that an optimal transport $T_i^\star$ (or, equivalently, the gradient of a convex potential $f_i^\star$) explains how measure $\mu_i$ was mapped to $\nu_i$, given context $c_i$; (ii.) we build on the multi-task hypothesis (Caruana, 1997) that all of the $N$ maps $T_i^\star$ between $\mu_i$ and $\nu_i$ share a common set of parameters, that are *modulated* by context informations $c_i$. These ideas are summarized in an abstract regression model described below.

### 3.1 A Regression Formulation for Conditional OT Estimation

$\theta \in \Theta \subset \mathbb{R}^r$, $\mathcal{T}_\theta$ describes a function that takes an input vector $c \in \mathcal{C}$, and outputs a *function* $\mathcal{T}_\theta(c) : \mathbb{R}^d \to \mathbb{R}^d$, as a hypernetwork would (Ha et al., 2016). Assume momentarily that we are given *ground truth* maps $T_i$, that describe the effect of context $c_i$ on any measure, rather only pairs of measures $(\mu_i, \nu_i)$. This is of course a major leap of faith, since even recovering an OT map $T^\star$ from two measures is in itself very challenging (Hütter and Rigollet, 2021; Rigollet and Stromme, 2022; Pooladian and Niles-Weed, 2021). If such maps were available, a direct supervised approach to learn a unique $\theta$ could hypothetically involve minimizing a fit function composed of losses between maps

$$\min_\theta \sum_{i=1}^N \int_{\mathbb{R}^d} \|\mathcal{T}_\theta(c_i)(x) - T_i(x)\|^2 \, \mathrm{d}\mu_i(x) \, . \tag{4}$$

Unfortunately, such maps $T_i$ are not given, since we are only provided unpaired samples before $\mu_i$ and after $\nu_i$ that map's application. By Brenier's theorem, we know, however, that such an OT map $T_i^\star$ exists, and that it would be necessarily the gradient of a convex potential function that maximizes (2). As a result, we propose to modify (4) to (i.) parameterize, for any $c$, the map $\mathcal{T}_\theta(c)$ as the gradient w.r.t. $x$ of a function $f_\theta(x, c) : \mathbb{R}^d \times \mathcal{C} \to \mathbb{R}$ that is convex w.r.t. $x$, namely $\mathcal{T}_\theta(c) := x \mapsto \nabla_1 f_\theta(x, c)$; (ii.) estimate $\theta$ by maximizing *jointly* the dual objectives (2) simultaneously for all $N$ pairs of measures, in order to ensure that the maps are close to optimal, to form the aggregate problem

$$\max_\theta \sum_{i=1}^N \mathcal{E}_{\mu_i, \nu_i}(f_\theta(\,\cdot\,, c_i)). \tag{5}$$

We detail in App. B how the Legendre transforms that appear in the energy terms $\mathcal{E}_{\mu_i, \nu_i}$ are handled with an auxiliary function.

## 3.2 Integrating Context in Convex Architectures

We propose to incorporate context variables, in order to modulate a family of convex functions $f_\theta(x, c)$ using partially input convex neural networks (PICNN). PICNNs are neural networks that can be evaluated over a pair of inputs $(x, c)$, but which are only required to be convex w.r.t. $x$. Given an input vector $x$ and context vector $c$, a $K$-layer PICNN is defined as $\psi_\theta(x, c) = z_K$, where, recursively for $0 \leq k \leq K - 1$ one has

$$u_{k+1} = \tau_k \left( V_k u_k + v_k \right),$$
$$z_{k+1} = \sigma_k \left( W_k^z \left( z_k \circ [W_k^{zu} u_k + b_k^z]_+ \right) + W_k^x \left( x \circ (W_k^{xu} u_k + b_k^x) \right) + W_k^u u_k + b_k^u \right), \tag{6}$$

where the PICNN is initialized as $u_0 = c, z_0 = \mathbf{0}$, $\circ$ denotes the Hadamard elementwise product, and $\tau_k$ is any activation function. The parameters of the PICNN are then given by

$$\theta = \{V_k, W_k^z, W_k^{zu}, W_k^x, W_k^{xu}, W_k^u, v_k, b_k^z, b_k^x, b_k^u\}.$$

Similar to ICNNs, the convexity w.r.t. input variable $x$ is guaranteed as long as activation functions $\sigma_i$ are convex and non-decreasing, and the weight matrices $W_k^z$ have non-negative entries. We parameterize this by storing them as elementwise applications of softplus operations on precursor matrices of the same size, or, alternatively, by regularizing their negative part. Finally, much like ICNNs, all matrices at the $K - 1$ layer are line vectors, and their biases scalars.

Such networks were proposed by Amos et al. (2017, Eq. 3) to address a problem that is somewhat symmetric to ours: Their inputs were labeled as $(y, x)$, where $y$ is a label vector, typically much smaller than that of vector $x$. Their PICNN is convex w.r.t. $y$, in order to easily recover, given a datapoint $x$ (e.g., an image) the best label $y$ that corresponds to $x$ using gradient descent as a subroutine, i.e. $y^\star(x) = \arg\min_y \text{PICNN}_\theta(x, y)$. PICNN were therefore originally proposed to learn a parameterized, implicit classification layer, amortized over samples, whose motivation rests on the property that it is convex w.r.t. label variable $y$. By contrast, we use PICNNs that are convex w.r.t. data points $x$. In addition to that swap, we do not use the convexity of the PICNN to define an implicit layer (or to carry out gradient descent). Indeed, it does not make sense in our setting to minimize $\psi_\theta(x, c)$ as a function of $x$, since $x$ is an observation. Instead, our work rests on the property that $\nabla_1 \psi_\theta(x, c)$ describes a parameterized family of OT maps. We note that PICNNs were considered within the context of OT in (Fan et al., 2021, Appendix B). In that work, PICNN provide an elegant reformulation for neural Wasserstein barycenters. Fan et al. (2021) considered a context vector $c$ that was restricted to be a small vector of probabilities.

## 3.3 Conditional Monge Map Architecture

Using PICNNs as a base module, the CONDOT architecture integrates operations on the contexts $\mathcal{C}$. As seen in Figure 1, context values $c$ may take various forms:

1. A scalar $t$ denoting a strength or a temporal effect. For instance, McCann's interpolation and its time parameterization, $\alpha_t = ((1 - t)\text{Id} + tT)_\sharp \alpha_0$ (McCann, 1997) can be interpreted as a trivial conditional OT model that creates, from an OT map $\psi_\theta$, a set of maps parameterized by $t$, $\mathcal{T}_\theta(t) := x \mapsto \nabla_x \left( (1 - t)\|x\|^2/2 + t\psi_\theta(x) \right)$.
2. A covariate vector influencing the nature of the effect that led $\mu_i$ to $\nu_i$, (capturing, e.g., patient feature vectors).
3. One or multiple actions, possibly discrete, representing decisions or perturbations applied onto $\mu_i$.

To provide a flexible architecture capable of modeling different types of conditions as well as conditions appearing in combinations, the more general CONDOT architecture consists of the

hypernetwork $\mathcal{T}_\theta$ that is fed a context vector through embedding and combinator modules. This generic architecture provides a one-size fits all approach to integrate all types of contexts $c$.

**Embedding Module**   To give greater flexibility when setting the context variable $c$, CONDOT contains an embedding module $\mathcal{E}$ that translates arbitrary contexts into real-valued vectors. Besides simple scalars $t$ (Fig. 1a) for which no embedding is required, discrete contexts can be handled with an embedding module $\mathcal{E}_\phi$. When the set $\mathcal{C}$ is small, this can be done effectively using one-hot embeddings $\mathcal{E}_{\text{ohe}}$. For more complicated actions $a$ such as treatments, there is no simple way to vectorize a context $c$. Similarly to action embeddings in reinforcement learning (Chandak et al., 2019; Tennenholtz and Mannor, 2019), we can learn embeddings for discrete actions into a learned continuous representation. This often requires domain-knowledge on the context values. For molecular drugs, for example, we can learn molecular representations $\mathcal{E}_{\text{mol}}$ such as chemical, motif-based (Rogers and Hahn, 2010) or neural fingerprints (Rong et al., 2020; Schwaller et al., 2022). However, often this domain knowledge is not available. In this work, we thus construct so-called *mode-of-action* embeddings, by computing an embedding $\mathcal{E}_{\text{moa}}$ that encourages actions $a$ with similar effect on target population $\nu$ to have a similar representation. In § 5, we analyze several embedding types for different use-cases.

**Combinator Module**   While we often have access to contexts $c$ in isolation, it is crucial to infer the effect of contexts applied in combination. A prominent example are cancer combination therapies, in which multiple treatment modalities are administered in combination to enhance treatment efficacy (Kummar et al., 2010). In these settings, the mode of operation between individual contexts $c$ is often not known, and can thus not be directly modeled via simple arithmetic operations such as `min`, `max`, `sum`, `mean`. While we test as a baseline the case, applicable to one-hot-embeddings, where simple additions are used to model these combinations, we propose to augment the CONDOT architecture with a parameterized combinator module $\mathcal{C}_\Phi$. If the order in which the actions are applied is irrelevant or unknown, the corresponding network $\mathcal{C}_\Phi$ needs to be permutation-invariant, which can be achieved

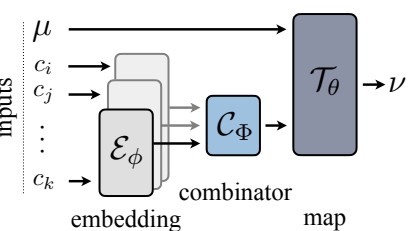

Figure 2: **CONDOT Architecture and Modules**. The embedding module $\mathcal{E}_\phi$ embeds arbitrary conditions $c$, which are then combined via module $\mathcal{C}_\Phi$. Using the processed contexts $c$, the map $\mathcal{T}_\theta(c)$ acts on $\mu$ to predict the target measure $\nu$.

by using a deep set architecture (Zaheer et al., 2017). Receiving a flexible number of inputs from the embedding module $\mathcal{E}_\phi$, CONDOT allows for a joint training of the PICNN parameters $\theta$, embedding parameters $\phi$, and combinator parameters $\Phi$ in a single, end-to-end differentiable architecture.

**Training Procedure**   Given a dataset $\mathcal{D} = \{c_i, (\mu_i, \nu_i)\}_{i=0}^N$ of $N$ pairs of populations before $\mu_i$ and after transport $\nu_i$ connected to a context $c_i$, we detail in Algorithm 1 provided in § B, a training loop that incorporates all of the architecture proposals described above. The training loss aims at making sure the map $\mathcal{T}_\theta(c_i)$ is an OT map from $\mu_i$ to $\nu_i$, where $c_i$ may either be the original label itself or its embedded/combined formulation in more advanced tasks. To handle the Legendre transform in (2), we use the proxy dual objective defined in (Makkuva et al., 2020, Eq. 6) (15)-(16) in place of (2) in our overall loss (5). This involves training the CONDOT architecture using two PICNNs, i.e., PICNN$_{\theta_f}$ and PICNN$_{\theta_g}$, that share the same embedding/combinator module, with a regularization (14) promoting that for any $c$, the PICNN$_{\theta_g}(\cdot, c)$ resembles the Legendre transform of the other, PICNN$_{\theta_f}^*(\cdot, c)$.

## 4  Initialization Strategies for Neural Convex Architectures

We address the problem of initializing the parameters of (P)ICNNs to ensure their gradient evaluated at every point is (initially) meaningful in the context of OT, namely that it is able to map the first and second moments of a measure $\mu$ into those of a target measure $\nu$. The initializers we propose build heavily on the quadratic layers proposed in the seminal reference (Korotin et al., 2020, Appendix B.2), notably the "DenseQuad" layer, as well as on closed-form solutions available for Gaussian approximations of measures (Gelbrich, 1990).

**Closed-Form Potentials for Gaussians**   Given two Gaussian distributions $\mathcal{N}_1, \mathcal{N}_2$ with means respectively $\mathbf{m}_1, \mathbf{m}_2$ and covariance matrices $\Sigma_1, \Sigma_2$ (where $\Sigma_1$ is assumed to be full rank), the

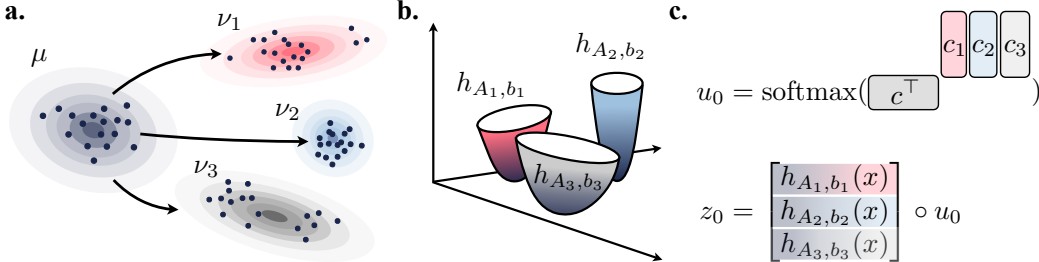

Figure 3: **a.** From a measure $\mu$ to several target measures $\nu_1, \nu_2, \nu_3$ provided with labels $c_1, c_2, c_3$ we can extract three Gaussian (quadratic) potentials in closed form, **b.** whose gradients transport on a first approximation $\mu$ to areas in space that cover the three targets. **c.** Given a new label vector $c$, we compare it to known labels to modulate the magnitude of each of the three potentials.

Brenier potential solving the OT problem from the first to the second Gaussian reads:

$$f^\star_{\mathcal{N}_1, \mathcal{N}_2} = \tfrac{1}{2} x^T A^T A x + b^T x + t(A, b) = \tfrac{1}{2}\|Ax\|_2^2 + b^T x + t(A, b), \text{ where,} \qquad (7)$$

$$A := \left( \Sigma_1^{-1/2} \left( \Sigma_1^{1/2} \Sigma_2 \Sigma_1^{1/2} \right)^{1/2} \Sigma_1^{-1/2} \right)^{1/2}, \quad b := \mathbf{m}_2 - A^T A \mathbf{m}_1,$$

define both quadratic and linear terms and $t(A, b)$ can be any constant. Importantly, note that we write the quadratic term in factorized form $AA^T$ to enforce psd-ness, as done by Korotin et al. (2020), not as usually done with a single psd matrix (Peyré and Cuturi, 2019, Remark 2.31).

Our quadratic potentials are only injected in the first state of hidden vector $z_0$, to populate it with a collection of relevant full-rank quadratic convex functions, with the goal of recovering an affine OT map from the start, as illustrated in the experiments from § C.1.

**Quadratic Potentials Lower Bounded by 0**   Naturally, for any choice of $t(A, b)$ one recovers the property that $\nabla f^\star_{\mu,\nu} \sharp \mathcal{N}_1 = \mathcal{N}_2$. When used in deep architectures, the level of that constant does, however, play a role, since convex functions in ICNN are typically thresholded or modulated using rectifying functions. To remove this ambiguity, we settle on a choice for $t(A, b)$ that is such that the lowest value reached by $f^\star_{\mathcal{N}_1, \mathcal{N}_2}$ is 0. This can be obtained by setting

$$t(A, b) := b^T (A^T A)^{-1} b, \qquad (8)$$

which results in the following choice, writing $\omega = \mathbf{m}_1 - (A^T A)^{-1} \mathbf{m}_2$,

$$f^\star_{\mathcal{N}_1, \mathcal{N}_2}(x) = \tfrac{1}{2}\|A \left( x + (A^T A)^{-1} b \right)\|_2^2 = \tfrac{1}{2}\|A \left( x - \omega \right)\|_2^2. \qquad (9)$$

To mimic these potential functions, we introduce a quadratic *layer* parameterized by a weight matrix $M$ and a "bias" vector $m$, defined as $q_{M,m}(x) = \tfrac{1}{2}\|M(x - m)\|_2^2$. By design, $q_{M,m}(x)$ is a convex quadratic, non-negative layer. Finally, one has the following relationships,

$$\nabla q_{I, \mathbf{0}_d} = \text{Id}, \quad \nabla q_{A, \omega} \sharp \mathcal{N}_1 = \mathcal{N}_2. \qquad (10)$$

**ICNN Initialization**   We explore two possible ICNN (3) initializers for OT.

Identity Initialization The first approach ensures that upon initialization the ICNN's gradient mimics the *identity* map, i.e., $\nabla \psi_\theta(x) = x$ for any $x$. We do so by injecting in the initial hidden state $z_0$ the norm of the input vector $\tfrac{1}{2}\|x\|^2$, cast as a trainable layer $q_{M,m}$ initialized with $M = I$ and $m = \mathbf{0}_d$, see (10). The remaining parameters are chosen to propagate that norm throughout layers using averages. This amounts to the following choices:

1. Set all $\sigma_i$ to be activations such that $\sigma_i'(u) \approx 1$ for $u$ large enough, e.g., (leaky) ReLU or softplus.
2. Introduce an initialization layer, $z_0 = q_{M,m}(x)\mathbf{1}$, itself initialized with $M = I$ and $m = \mathbf{0}_d$.
3. Initialize all matrices $W_i^z$ to $\approx \mathbf{1}_{d_2, d_1}/d_1$, where $d_1, d_2$ are the dimensions of these matrices.
4. Initialize all matrices $W_i^x$ to $\approx 0$.
5. Initialize biases $b_i$ to $s\mathbf{1}$, where $s$ is a large enough value $s$ so that $\sigma_i'(s) \approx 1$.

Gaussian Initialization The second approach can be used to initialize an ICNN so that its gradient mimics the affine transport between the Gaussian approximations of $\mu$ and $\nu$. To this end, we follow all of the steps outlined above, except for step 2 where the quadratic layer $q_{M,m}$ is initialized instead with $M = A$ and $m = \mathbf{m}_1 - (A^T A)^{-1} \mathbf{m}_2$ using notations in (7), (8), (9), where $\mathbf{m}_1, \mathbf{m}_2, \Sigma_1, \Sigma_2$

Table 1: Evaluation of drug effect predictions from control cells to cells treated with drug Givinostat when conditioning on various covariates influencing cellular responses such as drug dosage and cell type. Results are reported based on MMD and the $\ell_2$ distance between perturbation signatures of marker genes in the 1000 dimensional gene expression space.

| Method | Conditioned on Drug Dosage | | | | Conditioned on Cell Line | |
|---|---|---|---|---|---|---|
| | In-Sample | | Out-of-Sample | | In-Sample | |
| | MMD | $\ell_2$(PS) | MMD | $\ell_2$(PS) | MMD | $\ell_2$(PS) |
| CPA (Lotfollahi et al., 2021) | $0.1502 \pm 0.0769$ | $2.47 \pm 2.89$ | $0.1568 \pm 0.0729$ | $2.65 \pm 2.75$ | $0.2551 \pm 0.006$ | $2.71 \pm 1.51$ |
| ICNN OT (Makkuva et al., 2020) | $0.0365 \pm 0.0473$ | $2.37 \pm 2.15$ | $0.0466 \pm 0.0479$ | $2.24 \pm 2.39$ | $0.0206 \pm 0.0109$ | $1.16 \pm 0.75$ |
| CONDOT (Identity initialization) | $0.0111 \pm 0.0055$ | $0.63 \pm 0.09$ | $0.0374 \pm 0.0052$ | $2.02 \pm 0.10$ | $0.0148 \pm 0.0078$ | $0.39 \pm 0.06$ |
| CONDOT (Gaussian initialization) | $0.0128 \pm 0.0081$ | $0.60 \pm 0.11$ | $0.0325 \pm 0.0062$ | $1.84 \pm 0.14$ | $0.0146 \pm 0.0074$ | $0.41 \pm 0.07$ |

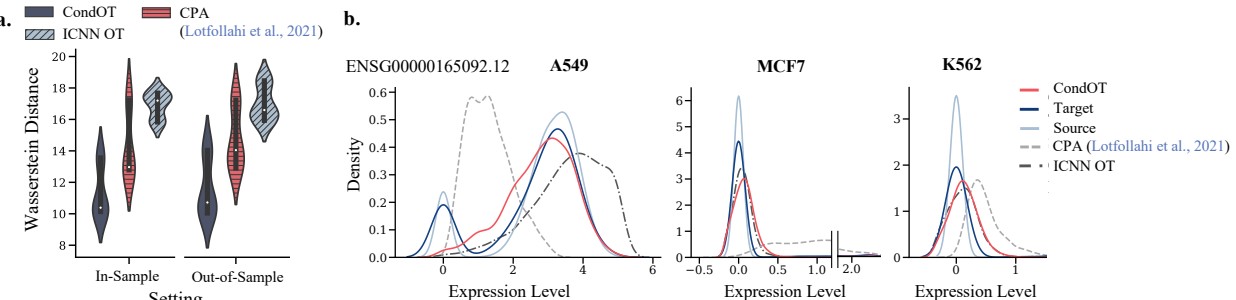

Figure 4: **a.** Predictive performance of CONDOT and baselines w.r.t. the entropy-regularized Wasserstein distance on drug dosages *in-sample*, i.e., seen during training, and *out-of-sample*, i.e., unseen during training. **b.** Marginal distributions of observed source and target distributions, as well as predictions on perturbed distributions by CONDOT and baselines of an exemplary gene across different cell lines. Predicted marginals of each method should match the marginal of the target population.

are replaced by the empirical mean and covariances of $\mu$ and $\nu$. Throughout the experiments, we use the Gaussian and identity initialization. Further comparisons between the vanilla initialization and those introduced in this work can be found in § C.1 (Fig. 8).

**PICNN Initialization**  Recall for convenience that a $K$-layer PICNN architecture reads:

$$u_{k+1} = \tau_k \left( V_k u_k + v_k \right)$$
$$z_{k+1} = \sigma_k \left( W_k^z \left( z_k \circ [W_k^{zu} u_k + b_k^z]_+ \right) + W_k^x \left( x \circ (W_k^{xu} u_k + b_k^x) \right) + W_k^u u_k + b_k^u \right)$$
$$\psi_\theta(x, c) = z_K.$$

In their original form (Amos et al., 2017, Eq. 3), PICNNs are initialized by setting $u_0 = c$ and $z_0 = \mathbf{0}$ to a zero vector of suitable size. Intuitively, the hidden states $u_k$ act as context-dependent modulators, whereas vectors $z_k$ propagate, layer after layer, a collection of convex functions in $x$ that are iteratively refined, while retaining the property that they are each convex in $x$. A reasonable initialization for a PICNN that is provided a context vector $c$ is that if $c \approx c_j$ (where $j$ is in the training set), one has that $\nabla_1 \psi_{\theta_0}(\cdot, c)$ maps approximately $\mu_j$ to $\nu_j$, which can be obtained by having $\psi_{\theta_0}(\cdot, c)$ mimic the closed-form Brenier potential between the Gaussian approximations of $\mu_j, \nu_j$. Alternatively, one may also default to an identity initialization as discusses above. To obtain either behavior, we make the following modifications, and refer to the illustration in Fig. 3:

1. The modulator $u_0(c) = \text{softmax}(c^T M)$, where $C$ is initialized as $M = [c_j]_j$, and $V_i = I, v_i = \mathbf{0}$.
2. $z_0 = [q_{M_j, m_j}(x)]_j$, where weight matrices and bias $(M_j, m_j)$ are either initialized to $(I, \mathbf{0})$ or as $(A_j, \omega_j)$ recovered by solving the Gaussian affine map from $\mu_j$ to $\nu_j$ using (9).
3. Modulator $u_0$ is passed directly to hidden state upon first iteration $W_0^{zu} = I, b_0^z = 0$.
4. All subsequent matrices $W_k^z$ are initialized to $\approx \mathbf{1}_{d_2, d_1}/d_1$, where $d_1, d_2$ are their dimensions,
5. $W_k^x$ and $W_k^{xu}$ are $\approx 0$, the biases $b_k^z \approx \mathbf{1}, b_k^u \approx \mathbf{0}$.

## 5  Evaluation

Biological cells undergo changes in their molecular profiles upon chemical, genetic, or mechanical perturbations. These changes can be measured using recent technological advancements in high-

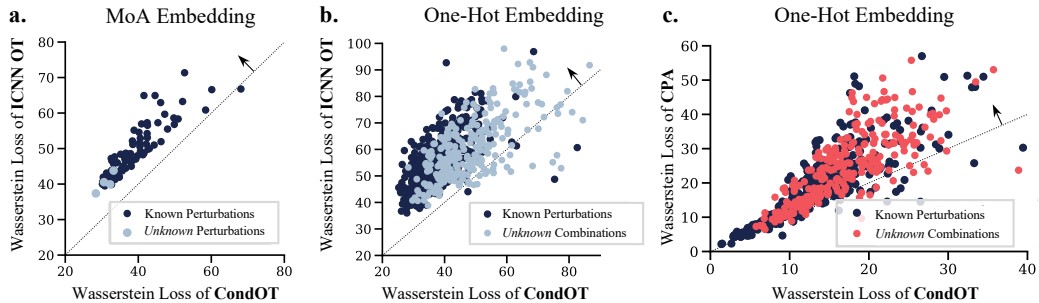

Figure 5: Comparison between **a.** CONDOT and ICNN OT (Makkuva et al., 2020) based on embedding $\mathcal{E}_{\mathrm{moa}}$ **b.** as well as $\mathcal{E}_{\mathrm{ohe}}$, and **c.** CONDOT and CPA (Lotfollahi et al., 2021) based on embedding $\mathcal{E}_{\mathrm{ohe}}$ on *known* and *unknown* perturbations or combinations. Results above the diagonal suggest higher predictive performance of CONDOT.

resolution multivariate single-cell biology. Measuring single cells in their unperturbed or perturbed state requires, however, to destroy them, resulting in populations $\mu$ and $\nu$ that are unpaired. The relevance of OT to that comes from its ability to resolve such ambiguities through OT maps, holding promises of a better understanding of health and disease. We consider various high-dimensional problems arising from this scenario to evaluate the performance of CONDOT (§ 3) versus other baselines.

## 5.1 Population Dynamics Conditioned on *Scalars*

Upon application of a molecular drug, the state of each cell $x_i$ of the unperturbed population is altered, and observed in population $\nu$. Molecular drugs are often applied at different dosage levels $t$, and the magnitude of changes in the gene expression profiles of single cells highly correlates with that dosage. We seek to learn a global, parameterized transport map $\mathcal{T}_\theta$ sensitive to that dosage. We evaluate our method on the task of inferring single-cell perturbation responses to the cancer drug Givinostat, a histone deacetylase inhibitor with potential anti-inflammatory, anti-angiogenic, and antineoplastic activities (Srivatsan et al., 2020), applied at different dosage levels, i.e., $t \in \{10\,\mathrm{nM}, 100\,\mathrm{nM}, 1,000\,\mathrm{nM}, 10,000\,\mathrm{nM}\}$. The dataset contains $3,541$ cells described with the gene expression levels of $1,000$ highly-variable genes. In a first experiment, we measure how well CONDOT captures the drug effects at different dosage levels via distributional distances such as MMD (Gretton et al., 2012) and the $\ell_2$-norm between the corresponding perturbation signatures (PS), as well as the entropy-regularized Wasserstein distance (Cuturi, 2013). We compute the metrics on 50 marker genes, i.e., genes mostly affected upon perturbation. For more details on evaluation metrics, see § E.2. To put CONDOT's performance into perspective, we compare it to current state-of-the-art baselines (Lotfollahi et al., 2021) as well as parameterized Monge maps without context variables (Bunne et al., 2021; Makkuva et al., 2020, ICNN OT), see § E.1. As visible in Table 1 and Fig. 4a, CONDOT achieves consistently more accurate predictions of the target cell populations at different dosage levels than OT approaches that cannot utilize context information, demonstrated through a lower average loss and a smaller variance. This becomes even more evident when moving to the setting where the population has been trained only on a subset of dosages and we test CONDOT on *out-of-sample* dosages. Table 1 and Fig. 4a demonstrate that CONDOT is able to generalize to previously *unknown* dosages, thus learning to interpolate the perturbation effects from dosages seen during training. For further analysis, we refer the reader to § E (see Fig. 9 and 10). We further provide an additional comparison of CONDOT, operating in the multi-task setting, to the single-task performance of optimal transport-based methods § C.4. While the single-task setting of course fails to generalize to new contexts and requires all contexts to be distinctly known, it provides us with a *pseudo* lower bound, which CONDOT is able to reach (see Table 2).

## 5.2 Population Dynamics Conditioned on *Covariates*

Molecular processes are often highly dependent on additional covariates that steer experimental conditions, and which are not present in the features measures in population $\mu$ or $\nu$. This can be, for instance, factors such as different cell types clustered within the populations. When the model can only

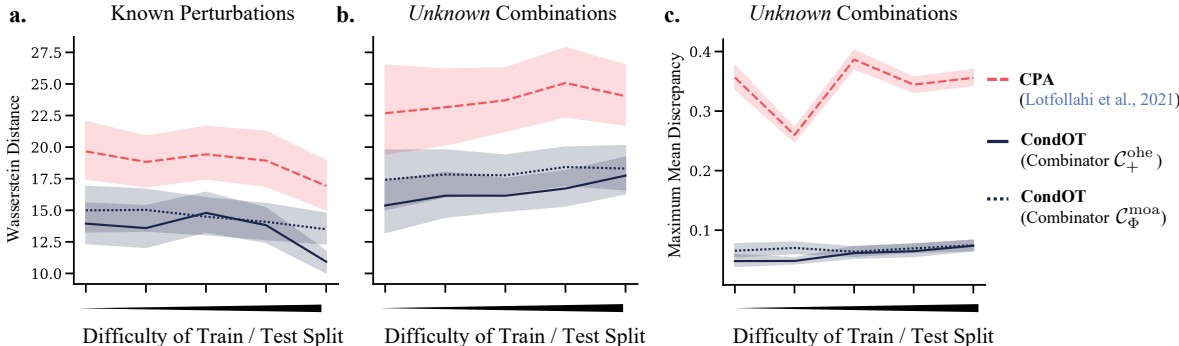

Figure 6: Predictive performance for **a.** known perturbations, **b.** unknown perturbations in combination w.r.t. regularized Wasserstein distance and **c.** MMD over different train / test splits of increasing difficulty for baseline CPA as well as CONDOT with different combinators $\mathcal{C}_+^{\text{ohe}}$ and $\mathcal{C}_\Phi^{\text{moa}}$. For more details on the dataset splits, see §D.2.

be conditioned w.r.t. a small and *fixed* set of metadata information, such as cell types, it is sufficient to encode these contexts using a one-hot embedding module $\mathcal{E}_{\text{ohe}}$. To illustrate this problem, we consider cell populations comprising three different cell lines (A549, MCF7, and K562). As visible in Table 1, CONDOT outperforms current baselines which equally condition on covariate information such as CPA (Lotfollahi et al., 2021), assessed through various evaluation metrics. Figure 4b displays a gene showing highly various responses towards the drug Givinostat dependent on the cell line. CONDOT captures the distribution shift from control to target populations consistently across different cell lines.

### 5.3 Population Dynamics Conditioned on *Actions*

To recommend personalized medical procedures for patients, or to improve our understanding of genetic circuits, it is key to be able to predict the outcomes of novel perturbations, arising from combinations of drugs or of genetic perturbations. Rather than learning individual maps $T_\theta^a$ predicting the effect of individual treatments, we aim at learning a global map $\mathcal{T}_\theta$ which, given as input the unperturbed population $\mu$ as well as the action $a$ of interest, predicts the cell state perturbed by $a$. Thanks to its modularity, CONDOT can not only learn a map $T_\theta$ for all actions *known* during training, but also to generalize to *unknown* actions, as well as potential *combinations* of actions. We will discuss all three scenarios below.

#### 5.3.1 *Known* Actions

In the following, we analyze CONDOT's ability to accurately predict phenotypes of genetic perturbations based on single-cell RNA-sequencing pooled CRISPR screens (Norman et al., 2019; Dixit et al., 2016), comprising $98,419$ single-cell gene expression profiles with 92 different genetic perturbations, each cell measured via a $1,500$ highly-variable gene expression vector. As, in a first step, we do not aim at generalizing beyond perturbations encountered during training, we utilize again a one-hot embedding $\mathcal{E}_{\text{ohe}}$ to condition $\mathcal{T}_\theta$ on each perturbation $a$. We compare our method to other baselines capable of modeling effects of a large set of perturbations such as CPA (Lotfollahi et al., 2021). Often, the effect of genetic perturbations are subtle in the high-dimensional gene expression profile of single cells. Using ICNN-parameterized OT maps without context information, we can thus assess the gain in accuracy of predicting the perturbed target population by incorporating context-awareness over simply predicting an average perturbation effect. Figure 5a and b demonstrate that compared to OT ablation studies, Fig. 5c and Fig. 6a for the current state-of-the-art method CPA (Lotfollahi et al., 2021). Compared to both, CONDOT captures the perturbation responses more accurately w.r.t. the Wasserstein distance.

#### 5.3.2 *Unknown* Actions

With the emergence of new perturbations or drugs, we aim at inferring cellular responses to settings not explored during training. One-hot embeddings, however, do not allow us to model *unknown* perturbations. This requires us to use an embedding $\mathcal{E}$, which can provide us with a representation of an unknown action $a'$. As genetic perturbations further have no meaningful embeddings as, for example,

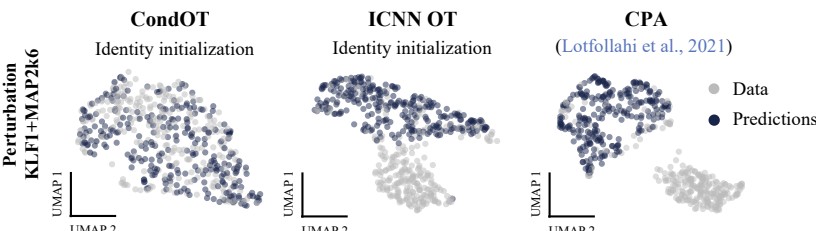

Figure 7: UMAP embeddings of cells perturbed by the combination KLF1+MAP2K6 (gray) and predictions of CONDOT (ours), ICNN OT (Makkuva et al., 2020), and CPA (blue). While CONDOT aligns well with observed perturbed cells, the baselines fail to capture subpopulations.

molecular fingerprints for drugs, we resort to mode-of-action embeddings introduced in § 3.3. Assuming marginal sample access to all individual perturbations, we compute a multidimensional scaling (MDS)-based embedding from pairwise Wasserstein distances between individual target populations, such that perturbations with similar effects are closely represented. For details, see § E. As current state-of-the-art methods are restricted to modeling perturbations via one-hot encodings, we compare our method to ICNN OT only. As displayed in Fig. 5a, CONDOT accurately captures the response of *unknown* actions (BAK1, FOXF1, MAP2K6, MAP4K3), which were not seen during training, at a similar Wasserstein loss as perturbation effects seen during training. For more details, see § E.

### 5.3.3 Actions in Combination

While experimental studies can often measure perturbation effects in biological systems in isolation, the combinatorial space of perturbations in composition is too large to capture experimentally. Often, however, combination therapies are cornerstones of cancer therapy (Mokhtari et al., 2017). In the following, we test different combinator architectures to predict genetic perturbations in combination from single targets. Similarly to Lotfollahi et al. (2021), we can embed combinations by adding individual one-hot encodings of single perturbations (i.e., $\mathcal{C}_+^{\text{ohe}}$). In addition, we parameterize a combinator via a permutation-invariant deep set, as introduced in § 3.3, based on mode-of-action embeddings of individual perturbations (i.e., $\mathcal{C}_\Phi^{\text{moa}}$). We split the dataset into train / test splits of increasing difficulty (details on the dataset splits in §D.2). Initially containing all individual perturbations as well as some combinations, the number of perturbations seen in combination during training decreases over each split. For more details, see § E. We compare different combinators to ICNN OT (Fig. 5b) and CPA (Lotfollahi et al., 2021) (Fig. 5c, Fig. 6b, c). While the performance drops compared to inference on *known* perturbations (Fig. 6a) and decreases with increasing difficulty of the train / test split, CONDOT outperforms all baselines. When embedding these high-dimensional populations in a low-dimensional UMAP space (McInnes et al., 2018), one can see that CONDOT captures the entire perturbed population, while ICNN OT and CPA fail in capturing certain subpopulations in the perturbed state (see Fig. 7 and 11).

## 6   Conclusion

We have developed the CONDOT framework that is able to infer OT maps from not only one pair of measures, but many pairs that come labeled with a context value. To ensure that CONDOT encodes optimal transports, we parameterize it as a PICNN, an input-convex NN that modulates the values of its weights matrices according to a sequence of feature representations of that context vector. We showcased the generalization abilities of CONDOT in the extremely challenging task of predicting outcomes for unseen combinations of treatments. These abilities and PICNN more generally hold several promises, both as an augmentation of the OTT toolbox (Cuturi et al., 2022), and for future applications of OT to single-cell genomics.

## Acknowledgments and Disclosure of Funding

This publication was supported by the NCCR Catalysis (grant number 180544), a National Centre of Competence in Research funded by the Swiss National Science Foundation. We thank Stefan Stark and Gabriele Gut for helpful discussions and the reviewers for their thoughtful comments and efforts towards improving our manuscript.

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
