# Supervised Training of Conditional Monge Maps

**Charlotte Bunne**[*]
ETH Zurich
bunnec@ethz.ch

**Andreas Krause**
ETH Zurich
krausea@ethz.ch

**Marco Cuturi**
Apple
cuturi@apple.com

## Abstract

Optimal transport (OT) theory describes general principles to define and select, among many possible choices, the most efficient way to map a probability measure onto another. That theory has been mostly used to estimate, given a pair of source and target probability measures $(\mu, \nu)$, a parameterized map $T_\theta$ that can efficiently map $\mu$ onto $\nu$. In many applications, such as predicting cell responses to treatments, pairs of input/output data measures $(\mu, \nu)$ that define optimal transport problems do not arise in isolation but are associated with a *context $c$*, as for instance a treatment when comparing populations of untreated and treated cells. To account for that context in OT estimation, we introduce CONDOT, a multi-task approach to estimate a family of OT maps conditioned on a context variable, using several pairs of measures $(\mu_i, \nu_i)$ tagged with a context label $c_i$. CONDOT learns a *global* map $\mathcal{T}_\theta$ conditioned on context that is not only expected to fit *all labeled pairs* in the dataset $\{(c_i, (\mu_i, \nu_i))\}$, i.e., $\mathcal{T}_\theta(c_i)\sharp\mu_i \approx \nu_i$, but should also *generalize* to produce meaningful maps $\mathcal{T}_\theta(c_{\text{new}})$ when conditioned on unseen contexts $c_{\text{new}}$. Our approach harnesses and provides a novel usage for *partially input convex neural networks*, for which we introduce a robust and efficient initialization strategy inspired by Gaussian approximations. We demonstrate the ability of CONDOT to infer the effect of an arbitrary combination of genetic or therapeutic perturbations on single cells, using only observations of the effects of said perturbations separately.

## 1 Introduction

A key challenge in the treatment of cancer is to predict the effect of drugs, or a combination thereof, on cells of a particular patient. To achieve that goal, single-cell sequencing can now provide measurements for individual cells, in treated and untreated conditions, but these are, however, not in correspondence. Given such examples of untreated and treated cells under different drugs, can we predict the effect of new drug combinations? We develop a general approach motivated by this and related problems, through the lens of *optimal transport (OT) theory*, and, in that process, develop tools that might be of interest for other application domains of OT. Given a collection of $N$ pairs of measures $(\mu_i, \nu_i)$ over $\mathbb{R}^d$ (cell measurements), tagged with a context $c_i$ (encoding the treatment), we seek to learn a context-dependent, parameterized transport map $\mathcal{T}_\theta$ such that, on training data, that map $\mathcal{T}_\theta(c_i) : \mathbb{R}^d \to \mathbb{R}^d$ fits the dataset, in the sense that $\mathcal{T}_\theta(c_i)\sharp\mu_i \approx \nu_i$. Additionally, we expect that this parameterized map can generalize to unseen contexts and patients, to predict, given a patient's cells described in $\mu_{\text{new}}$, the effect of applying context $c_{\text{new}}$ on these cells as $\mathcal{T}_\theta(c_{\text{new}})\sharp\mu$.

**Learning Mappings Between Measures** From generative adversarial networks, to normalizing flows and diffusion models, the problem of learning maps that move points from a source to a target distribution is central to machine learning. OT theory (Santambrogio, 2015) has emerged as a principled approach to carry out that task: For a pair of measures $\mu, \nu$ supported on $\mathbb{R}^d$, OT suggests that, among all maps $T$ such that $\nu$ can be reconstructed by applying $T$ to every point in the support

---

[*]Work done during an internship at Apple.

| **a.**     .. by scalar | **b.**     .. by covariate | **c.**     .. by action |
|---|---|---|
| e.g., time-course or dosage levels | e.g., metadata or identifiers | e.g., perturbations or decisions |

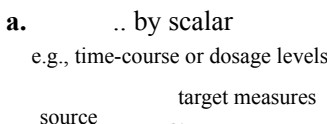
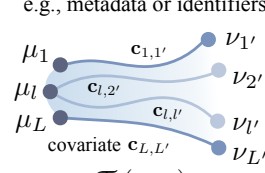
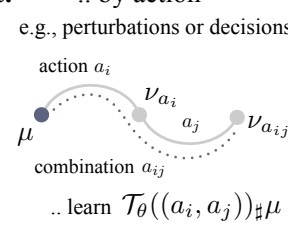

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

# Appendix

## A  Background

**Primal and Dual Optimal Transport**   The primal optimal transport problem (POT) was introduced in (1), and quickly linked in our background section § 2 to the dual optimal transport problem (DOT) (2). We provide for completeness an intermediary step to facilitate understanding, which works in the case where $p = 2$, and explain why the optimal transport map $T$ can also be recovered via the dual optimal transport problem. Introduced by Kantorovich in 1942, the dual formulation is a constrained concave maximization problem defined as

$$W(\mu, \nu) = \sup_{(f,g) \in \Psi_c} \int_{\mathcal{X}} f(x) \mathrm{d}\mu(x) + \int_{\mathcal{Y}} g(y) \mathrm{d}\nu(y),$$

where the set of admissible potentials is $\Psi_c := \{(f, g) \in L^1(\mu) \times L^1(\nu) : f(x) + g(y) \leq \frac{1}{2}\|x - y\|,$ $\forall(x, y)d\mu \otimes d\nu$ a.e.$\}$ (Villani, 2003, Theorem 1.3). The machinery of $c$-transforms (Santambrogio, 2015, §1.3 1) can be used to simplify that problem. When the cost $c$ is half the square Euclidean distance as considered here, this results in a simpler, so-called semi-dual problem (Cuturi and Peyré, 2018) that only involves a single potential function.

$$f^{\star}_{\mu,\nu} := \arg \sup_{f \text{ convex}} \int_{\mathbb{R}^d} f^* \mathrm{d}\mu + \int_{\mathbb{R}^d} f \mathrm{d}\nu = \psi^{\star}_{\mu,\nu}. \tag{11}$$

The optimal convex potential function $\psi$ is then related to the optimal dual potential $f^\star$ expressed above, through the identity $\psi = f^{\star}_{\mu,\nu}$.

**Neural Optimal Transport**   Learning optimal transport problems based on neural networks is at the core of many machine learning applications, including normalizing flows (Rezende and Mohamed, 2015; Huang et al., 2021) and generative models (Arjovsky et al., 2017; Genevay et al., 2019). Directly parameterizing the doubly-stochastic matrix $T$ of the primal optimal transport (1) as done in previous work (Jacob et al., 2018; Yang and Uhler, 2019; Prasad et al., 2020) has been shown to yield an unstable and thus difficult to solve optimization problem (Makkuva et al., 2020, Table 1). We thus follow previous work (Makkuva et al., 2020; Bunne et al., 2022b; Korotin et al., 2020; Alvarez-Melis et al., 2021) and instead learn map $T$ via the convex Brenier potential $\psi$ connected to the primal and dual optimal transport problem as outlined above. We parameterize the convex function $\psi$ via convex neural architectures (see § 2), which can thus be used in two contexts, either to model the Brenier potential, or to model a dual function. Both lead to the same results since the Brenier potential $\psi^{\star}_{\mu,\nu}$ is equal to the optimal dual potential associated with the second measure $\nu$, $f^{\star}_{\mu,\nu}$, as described above and in §2 around (2).

## B  The CONDOT Algorithm

CONDOT provides a generalized approach that from *labeled* pairs of measures $\{(c_i, (\mu_i, \nu_i))\}_i$ infers a *global* parameterized conditional Monge map $\mathcal{T}_\theta$. This is achieved by jointly learning an the embedding module $\mathcal{E}_\phi$, a combinator module $\mathcal{C}_\Phi$, as well as transport map $\mathcal{T}_\theta$. The algorithmic procedure is outlined in Algorithm 1. We describe CONDOT's modules as well as their parameterization in detail in §E.3. In the following, we will cover in more depth algorithmic approaches on how to learn transport map $\mathcal{T}_\theta$. Several approaches have been proposed on inferring transport map $\mathcal{T}_\theta$ from paired source and target populations, including the primal (1) or dual optimal transport problem (2).

A possible approach to learn our model could consist in minimizing a primal OT problem. In that case, we can learn $\mathcal{T}_\theta$ via the gradient of the Brenier potential parameterized via a PICNN, i.e., $\mathcal{T}_\theta = \nabla \psi^*_\theta = \nabla_1 \text{PICNN}_\theta$. The PICNN is then trained using the entropy-regularized Wasserstein distance (17) between the predictions $\hat{\nu} = \nabla \psi^*_\sharp \mu = \nabla_1 \text{PICNN}_\theta(\cdot, c)_\sharp \mu$ given source samples $\mu$ and condition $c$ and the observed target population $\nu$ as a loss function, i.e.,

$$\ell_{\text{POT}}(\mu, \nu, c; \theta) = W_\varepsilon(\nabla_1 \text{PICNN}_\theta(\cdot, c)_\sharp \mu, \nu). \tag{12}$$

Throughout this work, we choose a different route and propose instead to learn $\mathcal{T}_\theta$ via the dual optimal transport problem. We consider the strategy proposed by Makkuva et al. (2020) and utilized by Bunne et al. (2021) in the context of single-cell perturbation analyses. $\mathcal{T}_\theta$ is then parameterized via the pair of dual potentials $f$ and $g$, which themselves are defined by a pair of PICNNs $g : \text{PICNN}_{\theta_g}(\cdot, c)$

---

**Algorithm 1** CONDOT Algorithm.

---

**Input:** Dataset $\mathcal{D} = \{\mu_i, \nu_i, c_i\}_{i=0}^{N}$ of $N$ pairs of populations before $\mu_i$ and after transport $\nu_i$ connected to a context $c_i$, $\theta^0$ transport map $\mathcal{T}$ parameter initialization, $\phi^0$ embedding $\mathcal{E}$ parameter initialization, $\Phi^0$ embedding $\mathcal{C}$ parameter initialization, learning rates $\text{lr}_\theta$, $\text{lr}_\phi$ and $\text{lr}_\Phi$, and flag which `loss` function to use. In the case of the dual, we have $\theta = (\theta_f, \theta_g)$ parameterizing the dual potentials $f$ and $g$ and `train_freq_f` specifies the training frequency of dual potential $f$.

**Output:** Transport map $\mathcal{T}_\theta$, embedding $\mathcal{E}_\phi$, and combinator $\mathcal{C}_\Phi$.

1   $\theta, \phi, \Phi \leftarrow \theta^0, \phi^0, \Phi^0$
2   **for** $\{\mu_i, \nu_i, c_i\} \in \mathcal{D}$ **do**
     # Split (combination) context $c_i$ into individual contexts.
3     $c_i^1, c_i^2, \ldots, c_i^k = c_i$
4     $\hat{c}_i = \mathcal{C}_\Phi(\mathcal{E}_\phi(c_i^1), \mathcal{E}_\phi(c_i^2), \ldots, \mathcal{E}_\phi(c_i^k))$
5     **if** setting == 'dual' **then**
6       **if** i % train_freq_f == 0 **then**
7         $\ell \leftarrow \ell_{\text{DOT}}^{f}(\mu_i, \nu_i, \hat{c}_i; \theta_f)$   (15)
8       **else**
9         $\ell \leftarrow \ell_{\text{DOT}}^{g}(\mu_i, \nu_i, \hat{c}_i; \theta_g)$   (16)
10    **else**
11      $\ell \leftarrow \ell_{\text{POT}}(\mu_i, \nu_i, \hat{c}_i; \theta)$   (12)
     # Jointly optimize parameters $\theta, \phi, \Phi$ given loss $\ell$.
12    $\theta \leftarrow \theta - \text{lr}_\theta \times \nabla_\theta \ell$
13    $\phi \leftarrow \phi - \text{lr}_\phi \times \nabla_\phi \ell$
14    $\Phi \leftarrow \Phi - \text{lr}_\Phi \times \nabla_\Phi \ell$

15 **return**

---

and $f : \text{PICNN}_{\theta_f}(\cdot, c)$ such that $\hat{\nu} = \nabla g_\sharp \mu = \nabla_1 \text{PICNN}_{\theta_g}(\cdot, c) \sharp \mu$ is approximately $\nu$, as well as $\hat{\mu} = \nabla f_\sharp \nu = \nabla_1 \text{PICNN}_{\theta_f}(\cdot, c)_\sharp \nu$ is approximately $\mu$ on a labeled observation $((\mu, \nu), c)$ with parameters $\theta = (\theta_g, \theta_f)$. In order to optimize the pair of PICNNs, which parameterize the two dual functions, Makkuva et al. (2020) derive an approximate formulation of (2). First, Villani (2003, Theorem 2.9) rephrases (2) over the pair of dual potentials $(f, g)$ to

$$W(\mu, \nu) = \underbrace{\frac{1}{2}\mathbb{E}\left[\|x\| + \|y\|\right]}_{\mathcal{C}_{\mu,\nu}} - \inf_{f \text{ convex}} \mathbb{E}_\mu[f(X)] + \mathbb{E}_\nu\left[f^*(Y)\right], \tag{13}$$

where $f^*(y) = \sup_x \langle x, y \rangle - f(x)$ is $f$'s convex conjugate. In a second step, Makkuva et al. (2020) derive a min-max formulation by approximating the convex conjugate in (13) via

$$W(\mu, \nu) = \sup_{\substack{f \text{ convex} \\ f^* \in L^1(\nu)}} \inf_{g \text{ convex}} \mathcal{C}_{\mu,\nu} - \underbrace{\mathbb{E}_\mu[f(x)] - \mathbb{E}_\nu[\langle y, \nabla g(y) \rangle - f(\nabla g(y))]}_{\mathcal{V}_{\mu,\nu}(f,g)}, \tag{14}$$

and by relaxing the constraints on $g$. Thus, the dual potentials $f$ and $g$ can be learned via an alternate min-max optimization problem with loss functions

$$\ell_{\text{DOT}}^{f}(\mu, \nu, c; \theta_f) = \mathbb{E}_{x \sim \mu}[\text{PICNN}_{\theta_g}(x, c)] - \mathbb{E}_{y \sim \nu}[\text{PICNN}_{\theta_f}(\nabla \text{PICNN}_{\theta_g}(y, c), c)], \text{ and} \tag{15}$$

$$\ell_{\text{DOT}}^{g}(\mu, \nu, c; \theta_g) = -\mathbb{E}_{y \sim \nu}[\langle y, \nabla \text{PICNN}_{\theta_g}(y, c) \rangle - \text{PICNN}_{\theta_f}(\nabla \text{PICNN}_{\theta_g}(y, c), c)]. \tag{16}$$

For more details, see Makkuva et al. (2020); Korotin et al. (2021).

Thus, dependent on the strategy chosen, $\mathcal{T}_\theta$ is parameterized via a single or a pair of PICNN. Each network takes as input the source distribution $\mu$ —in which it is input convex— as well as an embedded context variable $\hat{c}$, returned by combinator $\mathcal{C}_\Phi$ and embedding module $\mathcal{E}_\phi$. Parameters of all three modules are jointly trained based on the derived optimal transport loss $\ell = \{\ell_{\text{POT}}, \ell_{\text{DOT}}\}$, which measures how close predicted target cells $\hat{\nu}$ are from the observed target population $\nu$, given source population $\mu$ and context $c$ as inputs.

# C Additional Experimental Results

## C.1 Comparison of Initialization Methods

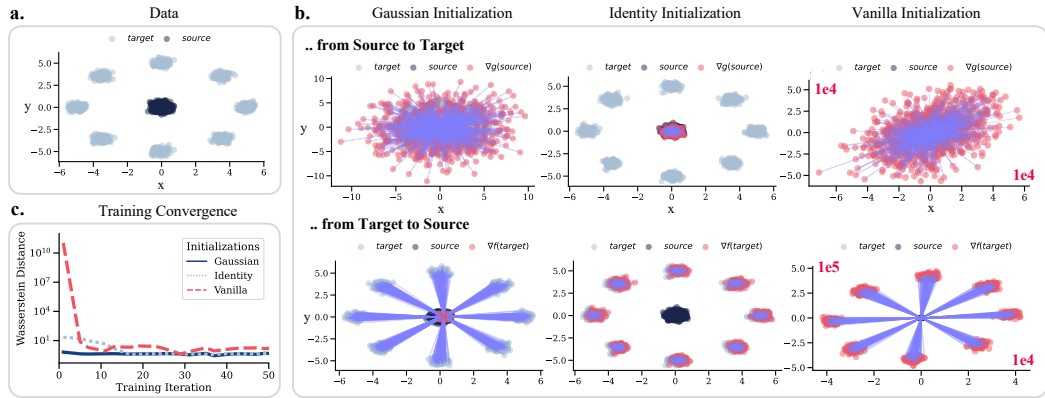

Figure 8: Comparison of ICNN initialization methods on a **a.** synthetic dataset containing source (dark blue) and target samples (light blue). **b.** Predicted samples (red) are obtained by transporting source samples (with dual potential $g$, first row) or target samples (with dual potential $f$, second row) to match the respective observations. The ICNNs are initialized such that they resemble a Gaussian closed-form approximation, the identity, or a random vanilla map (more details in § 4). Without any pretraining, the Gaussian initialization transports the samples to the Gaussian approximation of the respective target distribution. The identity initialization mimics the identity map and thus does not move the samples. The naïve vanilla initialization, on the other hand, starts with a solution far off from the target (i.e., values are in the range of $1e4$ or $1e5$). **c.** The chosen initialization strongly affects the convergence of the solution over the course of the training, here measured by the Wasserstein distance.

We conduct a simple experiment based on a synthetic dataset displayed in Fig. 8a, in which we seek to learn a mapping between source and target samples by parameterizing the dual potentials $f$ and $g$ with two ICNNs based on different initialization schemes (see Algorithm 1, § B). To showcase different initialization methods, we compare the initial predictions (at training iteration 0) of transported samples for the vanilla, the identity, and the Gaussian initialization (Fig. 8b). As the Gaussian initialization instantiates maps which transport source samples to the Gaussian approximation of the target samples, the initialization already captures well the source or target distribution using ICNN $f$ or $g$, respectively (Fig. 8b, first column). The identity initialization, instead, configures maps which do not move the samples from the initial distribution (see Fig. 8b, second column). Both initialization schemes proposed in this work thus result in map parameterizations, which initially (before training) realize non-trivial and admissible Monge maps. The vanilla initialization, on the other hand, instantiates random maps, mapping the point far away from the source and target distribution, thus impeding fast and robust training (see Fig. 8, third column). We want to stress that this is achieved without costly and elaborate pretraining of the networks as proposed in Korotin et al. (2020, Appendix B).

The selected initialization method also strongly affects the convergence of the solution over the course of the training, which we monitor using the Wasserstein distance between observed and predicted target samples (see Fig. 8c). In this simple example, the Gaussian initialization is already close to the solution, thus the Wasserstein distance resembles the final solution already at the beginning of the training. The identity initialization similarly starts with a mapping closer to the solution compared to a random vanilla initialization, thus achieving fast and stable convergence of the min-max optimization problem (14). This experiment thus demonstrates that a proper initialization is not only crucial for fast convergence but also the overall robustness of training and result.

## C.2    Out-of-Sample Predictions in Unknown Contexts

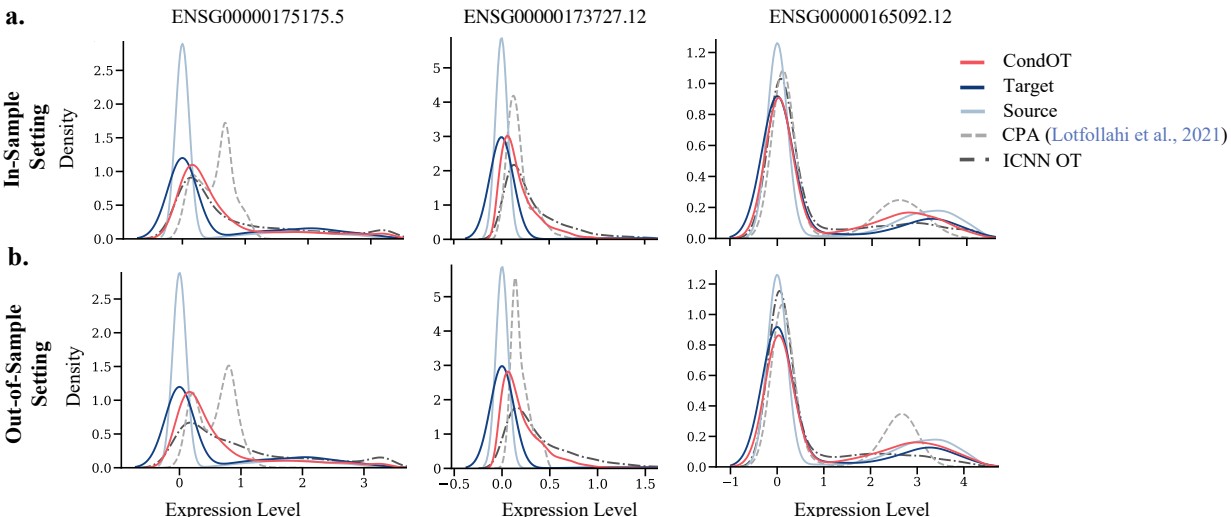

Figure 9: Marginal distributions of observed source (light blue) and target distributions (dark blue), as well as predictions on perturbed distributions by CONDOT (red) and baselines (gray) of different genes **a.** in the in-sample setting, where dosage 100nM was seen during training, and **b.** out-of-sample setting, where dosage 100nM was *not* seen during training. Predicted marginals of each method should match the marginal of the target population (dark blue. While the performance of CONDOT is consistent from the in-sample to the out-of-sample setting, both baselines show differences. These differences are subtle, however, only 3 out of $1,000$ genes are displayed.

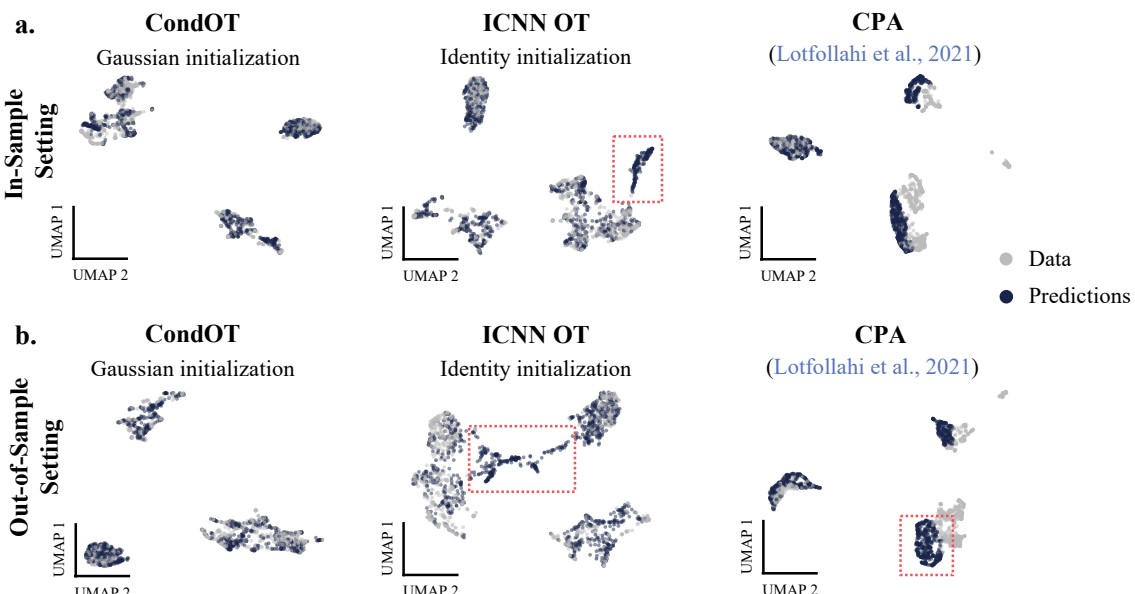

Figure 10: UMAP embeddings of cells perturbed by Givinostat dosage 100nM (gray) and predictions of CONDOT (ours), ICNN OT (Makkuva et al., 2020), and CPA (Lotfollahi et al., 2021) (blue). Contrary to the out-of-sample setting, the dosage 100nM was seen during training in the in-sample setting. While CONDOT covers the space of observed perturbed cells, the baselines fail to capture subpopulations (see red squares).

## C.3 Predicting Unknown Perturbations and Perturbations in Combination

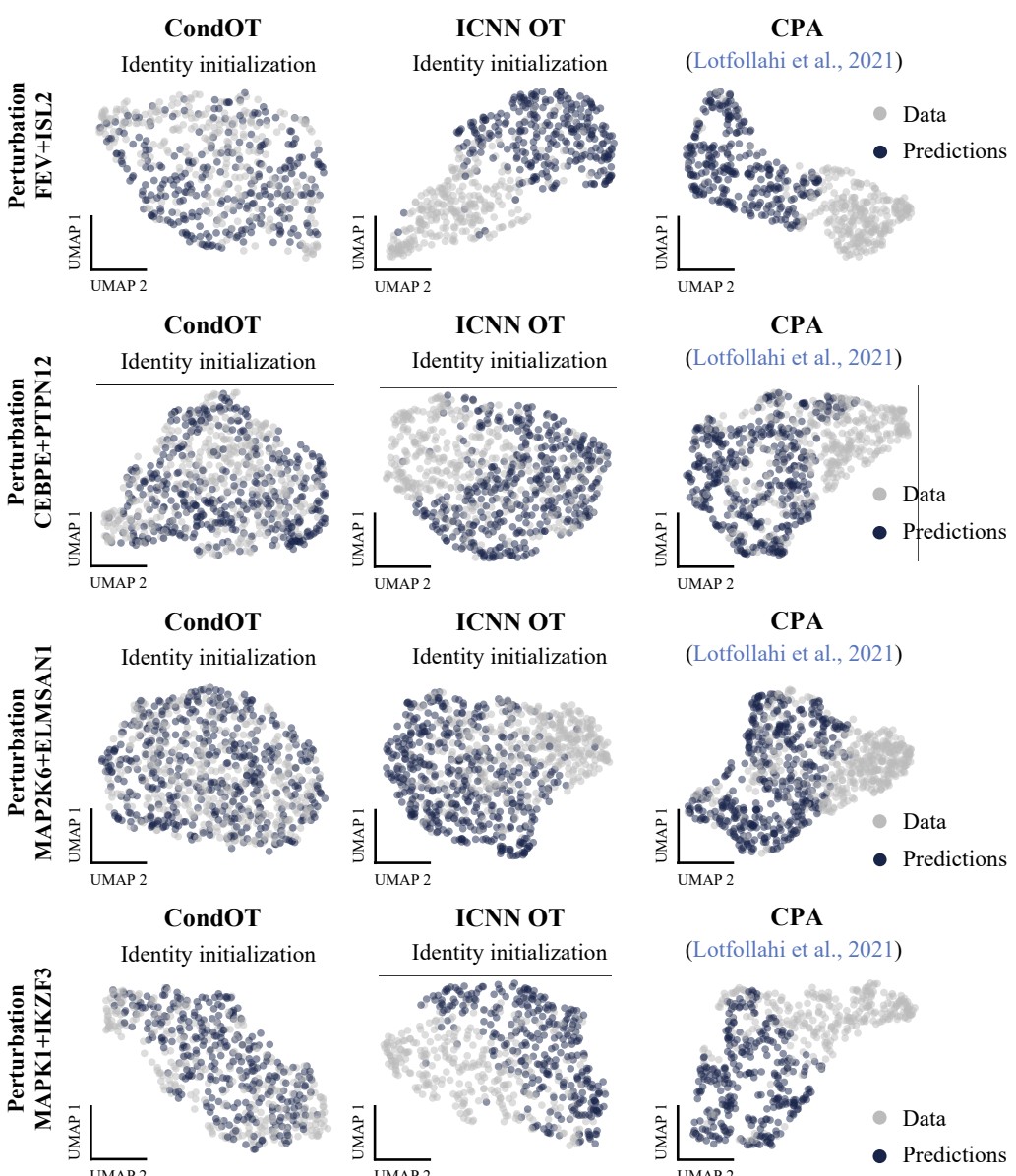

Figure 11: UMAP embeddings of cells perturbed by different combinations (grey) and predictions of CONDOT (ours), ICNN OT (Makkuva et al., 2020), and CPA (Lotfollahi et al., 2021) (blue). While CONDOT covers the space of observed perturbed cells, the baselines fail to capture subpopulations.

## C.4 Comparing Multi-Task Performance to Ideal Single-Task Baseline

In § 5.1 and 5.2 (Table 1), we compared CONDOT with different initializations against current state-of-the-art methods (Lotfollahi et al., 2021, CPA) and previous neural optimal transport-based methods used on single-cell data (Makkuva et al., 2020, ICNN OT). To challenge the performance of CONDOT even further, we add another baseline in which we train ICNN OT individually for each distinct condition. This baseline can be seen as a *lower bound* on what accuracy CONDOT can reach based on modeling perturbation responses by parameterizing Monge maps. In particular, we train CONDOT and both baselines (ICNN OT and CPA) to predict the dosage-dependent perturbation response to two different drugs, Trametinib and Givinostat. While both CONDOT and CPA allow

Table 2: Wasserstein loss $W_\varepsilon$ of different methods on the top-50 marker genes for the drugs Givinostat and Trametinib, where we conduct the analysis for different dosages individually on the dataset by Srivatsan et al. (2020).

| Dosages Model | Wasserstein Loss $W_\varepsilon$ | | | |
|---|---|---|---|---|
| | 10 nM | 100 nM | 1,000 nM | 10,000 nM |
| CPA (Lotfollahi et al., 2021) | $13.75 \pm 1.41$ | $13.75 \pm 0.93$ | $15.81 \pm 2.16$ | $55.12 \pm 58.14$ |
| ICNN OT (Makkuva et al., 2020) (on all conditions) | $12.37 \pm 1.66$ | $12.53 \pm 2.40$ | $13.36 \pm 3.02$ | $31.02 \pm 28.12$ |
| ICNN OT (Makkuva et al., 2020) (on selected condition) | $10.98 \pm 1.15$ | $10.37 \pm 0.23$ | $10.58 \pm 1.93$ | $20.55 \pm 14.16$ |
| CONDOT (Identity initialization) | $10.54 \pm 0.37$ | $10.58 \pm 0.04$ | $12.13 \pm 2.43$ | $20.97 \pm 15.02$ |
| CONDOT (Gaussian initialization) | $10.56 \pm 0.45$ | $10.54 \pm 0.16$ | $12.12 \pm 2.26$ | $21.30 \pm 15.29$ |

us to condition the training on all respective dosages, we train two variants of ICNN OT: The first version is trained on all conditions, while the additional baseline (the lower bound, i.e., ICNN OT selected condition) computes different and independent ICNN OT models for each dosage. While this would fail to generalize to new contexts and it requires all contexts to be distinctly known, this is, in a way, the best we can expect to achieve. We believe the setting in which we condition on scalars is a good start because in this 1D setting for $c$, the inability to generalize is less critical (as opposed to predicting previously unobserved combinations of drugs).

The results on on 50 marker genes in data space with 1000 genes are displayed in the Table 2. This additional experiment clearly demonstrates that CONDOT predicts perturbation responses as well as a baseline which was trained purely on individual conditions, while still being able to generalize (see Table 1). As often mentioned in the multitask learning literature (Mahabadi et al., 2021), sharing of parameters (the PICNN) and conditioning seems to improve by increasing the effective sample size of the problem.

# D  Datasets

We evaluate CONDOT on different tasks, consisting of a pair of source $\mu$ and target measures $\nu$, as well as context variables $c$ of different nature. In particular, we consider single-cell datasets in which populations of single cells have been monitored with modern high-throughput methods such as single-cell RNA sequencing technologies. Characterizing and modeling perturbation responses at the level of single cells with access to *unpaired* populations of control and perturbed cells remains one of the grand challenges of biology. In this work, we consider the task of modeling molecular responses to cancer drugs with context variables being the drug's dosage (i.e., a scalar, § 5.1) as well as covariates such as different cancer cell lines present in the population (§ 5.1). Further, we study cellular responses to genetic perturbations, where we condition on the perturbation, i.e., action, chosen. Here we differentiate between settings where we encounter *known* actions (§ 5.3.1), *unknown* actions (§ 5.3.2), and action applied in combination (§ 5.3.3) during evaluation. In the following, we introduce the datasets in more depth, describe preprocessing steps, feature selection, and data splits.

## D.1  ... by Srivatsan et al. (2020)

Cancer drugs reduce uncontrolled cell growth and proliferation by inhibiting DNA replication and RNA transcription as well as targeting proteins crucial for cancer progression. In doing so, they modulate downstream signaling cascades, affect cell growth and morphology, and alter gene expression profiles of single cells. Srivatsan et al. (2020) conduct a scRNA-seq–based phenotyping screen of transcriptional responses to thousands of independent perturbations at single-cell resolution. The measured cell population contains three well-characterized cancer cell lines, including A549, a human lung adenocarcinoma, K562, a chronic myelogenous leukemia, and MCF7, a mammary adenocarcinoma cell line. Due to different transcriptional profiles of each cancer cell line, drug compounds might cause divergent cellular responses in each subpopulation. For our analysis, we consider the drug Givinostat, a histone deacetylase inhibitor with potential anti-inflammatory, anti-angiogenic, and antineoplastic activities (Rambaldi et al., 2010). The dataset contains $17,565$ control cells as well as $3,541$ cells perturbed by Givinostat with different dosages, i.e., 10 nM, 100 nM, 1,000 nM, 10,000 nM.

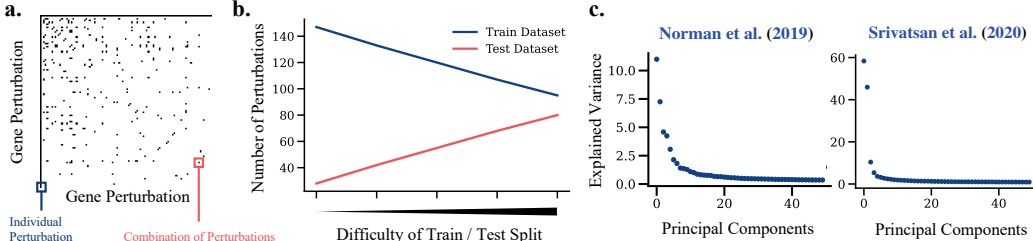

Figure 12: **a.** The indicator matrix of all individual perturbations as well as those perturbation pairs available in combination (black) in the dataset by Norman et al. (2019). **b.** Size of the different train / test splits of the dataset by Norman et al. (2019). The train set contains all single perturbations as well as a decreasing number of combinations with increasing difficulty of the data split. For more details, see §D.2. **c.** Explained variance of different datasets per principal component.

**Data Preprocessing**    The data is available for download in the Gene Expression Omnibus (GEO) database under accession number GSM4150378. For data quality control and preprocessing, we follow the analysis of Lotfollahi et al. (2021). The count matrix obtained from GEO consists of $581,777$ cells. The data was subset to half its size, with $290,888$ cells remaining after quality control for all $188$ different compounds. We proceeded with log-transformation and the the selection of $1,000$ highly-variant genes using `scanpy` (Wolf et al., 2018).

**Feature Selection**    Single-cell RNA sequencing data is very high-dimensional, even after selecting $1,000$ highly-variant genes. For the downstream analysis of how well the overall perturbation effect has been captured, we thus select the top $50$ marker genes, i.e., those genes which show strong differences between perturbed and unperturbed states. This analysis is conducted based on the `scanpy`'s function `rank_genes_groups`, setting unperturbed cells as reference (Wolf et al., 2018). It is important to note here, that CONDOT operates on the full dataset and the marker genes are only considered to report meaningful evaluation measures.

### D.2    ... by Norman et al. (2019)

Genetic interactions and their joint expression give rise to an inconceivable organismal complexity and uncountable many diverse phenotypes and behaviors. Constructing a systematic genetic interaction map is crucial for a better understanding of cellular mechanisms in health and disease. Thus, Norman et al. (2019) conducted single-cell, pooled transcriptional profiling of CRISPR-mediated perturbations to link genetic perturbation to its transcriptional consequences using the Perturb-Seq technology (Dixit et al., 2016). The dataset consists of individual perturbations as well as joint knockouts of different genes, allowing us to study the phenotypic consequences of perturbing a pair of genes alone or in combination. The indicator matrix of all individual perturbations as well as those pairs available in combination can be found in Fig. 12a.

**Data Preprocessing**    The data is available for download in the Gene Expression Omnibus (GEO) database under accession number GSE133344. For data quality control and preprocessing, we follow the analysis of Lotfollahi et al. (2021). We discarded those genetic perturbations with less than $250$ cells, resulting in a dataset with $92$ individual perturbations and $84$ perturbations in combination. This further included, the exclusion of particular subsets of control cells with in total $98,419$ remaining, data normalization, log-transformation, and selection of $1,500$ highly-variant genes using `scanpy` (Wolf et al., 2018).

**Feature Selection**    Similar as above, for evaluation we select the top $50$ marker genes, i.e., those genes most strongly affected by the particular genetic perturbation.

**Data Splits**    Following Lotfollahi et al. (2021), we create different train / test dataset splits of increasing difficulty. The train splits hereby always contain all $92$ individual perturbations as well as varying numbers of combinations. The easiest train split contains $55$ perturbations, while the test set only carries $28$ combinations which are unknown in the evaluation. Consecutive splits get

increasingly harder, comprising 42, 29, 16, and 4 combinations in the train set (besides all single perturbations) and 41, 54, 67, and 79 combinations in the test set, respectively (see Fig. 12b).

# E Experimental Details

In the following, we describe the experimental setup by providing an overview on the baselines, evaluation metrics, parameterizations of CONDOT's modules, and hyperparameters chosen.

## E.1 Baselines

We consider several baselines to put CONDOT's performance into perspective. This includes current state-of-the-art methods, as well as ablations of our methods.

**Compositional Perturbation Autoencoder (CPA)** Building up on previous work (Lotfollahi et al., 2019, 2020), the current state-of-the-art approach conditional perturbation autoencoder (CPA) learns transcriptional perturbation responses across different cell types, applied dosages, and perturbation combinations (Lotfollahi et al., 2021). The architecture hereby consists of several modules. CPA predicts perturbed states of populations by learning a factorized latent representation of both perturbations and covariates, with separate embeddings for particle feature vectors, perturbations, and external covariates. These embeddings are independent of each other by design to later allow modular recombination of different modules and thus allowing the model to make predictions on unseen perturbations in combination. We follow the experimental setup outlined in (Lotfollahi et al., 2021). Similarly to perturbations, covariates such as cell type or dosage are encoded via one-hot vectors. Thus, CPA can not be utilized to make predictions on *unknown* perturbations as studied in § 5.3.2.

**ICNN OT** A crucial ablation study of CONDOT is to learn the transition of source population $\mu$ to target population $\nu$ *without* considering context $c$. Thus, we use standard ICNNs (3) to parameterize the transport map module $\mathcal{T}_\theta$ via two dual potentials as proposed in Makkuva et al. (2020) and Bunne et al. (2021). As for the PICNNs, we utilize different initialization schemes as derived in § 4.

## E.2 Evaluation Metrics

Since we lack access to the ground truth pair of perturbed and unperturbed observations on the single cell level, we consider evaluation metrics on the level of the distribution of real and predicted perturbation states to analyze the effectiveness of CONDOT. We report results based on several metrics:

**Wasserstein Distance** We measure accuracy of the predicted target population $\hat{\nu}$ to the observed target population $\nu$ using the entropy-regularized Wasserstein distance (Cuturi, 2013) provided in the `OTT` library (Cuturi et al., 2022) defined as

$$W_\varepsilon(\hat{\nu}, \nu) := \min_{\mathbf{P} \in U(\hat{\nu}, \nu)} \langle \mathbf{P}, [\|x_i - y_j\|^2]_{ij} \rangle - \varepsilon H(\mathbf{P}), \tag{17}$$

where $H(\mathbf{P}) := -\sum_{ij} \mathbf{P}_{ij}(\log \mathbf{P}_{ij} - 1)$ and the polytope $U(\hat{\nu}, \nu)$ is the set of $n \times m$ matrices $\{\mathbf{P} \in \mathbb{R}_+^{n \times m}, \mathbf{P}\mathbf{1}_m = \hat{\nu}, \mathbf{P}^\top \mathbf{1}_n = \nu\}$. Throughout the evaluation, we set $\varepsilon = 0.1$.

**Maximum Mean Discrepancy** Kernel maximum mean discrepancy (Gretton et al., 2012) is another metric to measure distances between distributions, i.e., for our purpose between the predicted target population $\hat{\nu}$ to the observed target population $\nu$. Given two random variables $x$ and $y$ with distributions $\hat{\nu}$ and $\nu$, and a kernel function $\omega$, Gretton et al. (2012) define the squared MMD as:

$$\text{MMD}(\hat{\nu}, \nu; \omega) = \mathbb{E}_{x,x'}[\omega(x, x')] + \mathbb{E}_{y,y'}[\omega(y, y')] - 2\mathbb{E}_{x,y}[\omega(x, y)].$$

We report an unbiased estimate of $\text{MMD}(\hat{\nu}, \nu)$, in which the expectations are evaluated by averages over the population particles in each set. We utilize the RBF kernel, and as is usually done, report the MMD as an averaged over several length scales, i.e., $0.5, 0.1, 0.01$, and $0.005$.

**Perturbation Signatures** A common method to quantify the effect of a perturbation on a population is to compute its perturbation signature (Stathias et al., 2018, (PS)), computed via the difference in means between the distribution of perturbed states and control states of each feature, e.g., here individual genes. $\ell_2(\text{PS})$ then refers to the $\ell_2$-distance between the perturbation signatures computed

on the observed and predicted distributions, $\nu$ and $\hat{\nu}$. As before, let $\mu$ be the set of observed unperturbed population particles, $\nu$ the set of observed perturbed particles, as well as $\hat{\nu}$ the predicted perturbed state of population $\mu$. The $\ell_2$(PS) is then defined as

$$\text{PS}(\nu, \mu) = \frac{1}{m} \sum_{y_i \in \nu} y_i - \frac{1}{n} \sum_{x_i \in \mu} x_i,$$

where $n$ is the size of the unperturbed and $m$ of the perturbed population. We report the $\ell_2$ distance between the observed signature $\text{PS}(\nu, \mu)$ and the predicted signature $\text{PS}(\hat{\nu}, \mu)$, which is equivalent to simply computing the difference in the means between the observed and predicted distributions.

### E.3 CONDOT Modules

CONDOT consists of several modules for which different choices can be considered. Here, we provide a brief overview on the options and their parameterization.

#### E.3.1 Embedding Module $\mathcal{E}$

The embedding module allows us to consider context variables $c$ of various nature. In the case of scalars, no sophisticated embedding is necessary. In contrast, covariate contexts as well as potentially complex action descriptions require embeddings in order to be processed by the combinator $\mathcal{C}$, and transport map module $\mathcal{T}$.

**One-Hot Embedding $\mathcal{E}_{\mathbf{ohe}}$**     Covariates, such as subpopulation or patient identifiers, can be simply embedded via one-hot encodings. These embeddings, however, are not able to capture unknown covariates after training.

**Mode-of-Action Embedding $\mathcal{E}_{\mathbf{moa}}$**     In certain cases, actions might possess distinct properties which allow for a direct embeddings using this domain knowledge, i.e., molecular representations for molecules (Rong et al., 2020; Rogers and Hahn, 2010). In the case of genetic perturbations, however, no straightforward embedding is available. We thus introduce so-called mode-of-action embeddings, which map actions into a latent space-based on their mechanism of action and effect on the target population. In the fashion of word embeddings (Mikolov et al., 2013a,b,c), we require actions with similar effect to be closely embedded in the learned representation. This means, however, that we require some sample access of target population particles, i.e., perturbed cells by individual compounds (not in combination). While several metric embeddings are possible (Chopra et al., 2005), we here test a simple multi-dimensional scaling-based embedding (Mead, 1992). For this we compute the pairwise Wasserstein distance matrix between all target populations of different individual perturbations. We then compute a 10-dimensional MDS embedding-based on the stress minimization using majorization algorithm (smacof) (De Leeuw and Mair, 2009) of `sklearn` (Pedregosa et al., 2011), which serves as descriptor for each individual perturbation.

#### E.3.2 Combinator Module $\mathcal{C}$

The combinator module allows us to pass an arbitrary number of context $c$ to the transport map module $\mathcal{T}$.

**Multi-Hot Combinator $\mathcal{C}_{+}^{\mathbf{ohe}}$**     A naïve way of constructing the combinator is to combine different actions via a multi-hot encoding. If all single perturbations are observed during training, each individual action can be represented via a one-hot encoding. The potential combination of different actions, is then encoded by adding the respective one-hot encodings, resulting in a multi-hot encoding for each combination. A limitation of this embedding, however, is that it cannot generalize to unknown action after training.

**Deep Set Combinator $\mathcal{C}_{\Phi}^{\mathbf{moa}}$**     When not considering one-hot-based embeddings and when aiming to generalize to unseen perturbations, we need a combinator module which learns how to associate different individual embeddings with each other to receive a joint embedding. As we for now do not make an assumption on the order of the perturbation, we consider a permutation-invariance network architecture such as deep sets (Zaheer et al., 2017) with parameters $\Phi$. Taking a set of arbitrary size $k$ containing individual context embeddings $\{\mathcal{E}_{\text{moa}}(c^1), \mathcal{E}_{\text{moa}}(c^2), \dots, \mathcal{E}_{\text{moa}}(c^k)\}$, it returns a learned combination embedding $\hat{c}_i = \mathcal{C}_{\Phi}(\mathcal{E}_{\text{moa}}(c^1), \mathcal{E}_{\text{moa}}(c^2), \dots, \mathcal{E}_{\text{moa}}(c^k))$.

### E.3.3 Transport Map Module $\mathcal{T}$

The transport map module takes as input samples of the source distribution $\mu$ as well as context $c$ and returns the perturbed population $\nu$. Map $\mathcal{T}_\theta$ is thereby parameterized via PICNNs as we require input convexity in $\mu$ but not $c$. In the case where we consider learning $\mathcal{T}_\theta$ via the dual (2), it is defined by a pair of PICNNs with parameters $\theta = (\theta_f, \theta_g)$, parameterizing the set of dual variables $f$ and $g$. When deploying the primal OT problem (1), we parameterize a single Brenier potential via a PICNN with parameters $\theta$.

As suggested by Makkuva et al. (2020), we relax the convexity constraint on PICNN $g$ and instead penalize its negative weights $W_k^z$

$$R(\theta) = \lambda \sum_{W_k^z \in \theta} \|\max(-W_k^z, 0)\|_F^2.$$

The convexity constraint on PICNN $f$ is enforced after each update by setting the negative weights of all $W_k^z \in \theta_f$ to zero. Thus, the full objective then states

$$\max_{\theta_f : W_k^z \geq 0, \forall k} \min_{\theta_g} f_{\theta_f}(\nabla g_{\theta_g}(y)) - \langle y, \nabla g_{\theta_g}(y) \rangle - f_{\theta_f}(x) + \lambda R(\theta_g).$$

### E.3.4 Projection Module

For very high-dimensional inputs such as single-cell RNA seq data, we project the data into a lower-dimensional space. The effect of a perturbation effect is then learned on the control particles encoded into a lower dimensional space. Subsequently, we decode the predicted target particles into the original data space. We consider both, principal component (PCA) as well as autoencoder-based projections. When conducting experiments in PCA space, we consider the first 50 principal components, as they contain $> 99\%$ of the explained variance (see Fig. 6c). The autoencoder architecture is inspired by (Lotfollahi et al., 2019), as it has been designed and tested for single-cell RNA seq data. The results reported in § 5 are based on autoencoder projections and the evaluation metrics are computed on the decoded target particles.

## E.4 Hyperparameters

To learn the optimal transport maps, we use PICNN architectures of 4 hidden layers of width 64. The autoencoder parameterizing the projection module consists of an encoder and decoder with each 2 layers of 512 dimensional hidden layers. The size of the latent space is 50. The deep set consists of an encoder with 2 linear layers with 8 hidden units, followed by a `sum`-pooling operator and a 2 layer decoder with 8 hidden units, returning a set embedding of the same size as each individual input embedding, and passed through a final sigmoid activation function. For all networks, we use the Adam optimizer (Kingma and Ba, 2014) with a learning rate of 0.0001 ($\beta_1 = 0.5$, $\beta_2 = 0.9$) and $\lambda$=1. If $\mathcal{T}$ is learned via the OT dual, $f$ and $g$ are learned via an alternate min-max optimization. $f$ is updated by fixing $g$ and maximizing (15) with a single iteration. Then, for 10 iterations, i.e., `train_freq_f`$= 10$, $f$ is fixed, and $g$ is optimized by minimizing (16). For the baselines, we followed the default configurations specified by the authors on the same datasets. We use a default batch size of 256, which is adapted for perturbations with fewer cells (due to a train / test split of $80\%/20\%$).

## F Reproducibility

An implementation of CONDOT is available at github.com/bunnech/condot.