# OpenReview forum: "Supervised Training of Conditional Monge Maps"
_NeurIPS.cc/2022/Conference — NeurIPS 2022 Accept_

### Official Review · Reviewer_kDhV · 2022-06-30

**Rating:** 6
**Confidence:** 3
**Soundness:** 3 good
**Presentation:** 2 fair
**Contribution:** 2 fair

**Summary:**

This paper proposes to learn a global map that is conditioned on the contexts. They achieve this by minimizing the Wasserstein-2 distance between the pushforward distribution and target distribution. They assume the measures from source and target distributions are paired and parameterize their map as PICNN. Their contexts are quite flexible, including scalars, covariates, and discrete actions. To integrate the contexts, they design an embedding module and combinator module to help the map learn the conditional dependence. Depending on the complexity of the contexts, the embedding module can take one-hot embedding or multi-dimensional scaling embedding. The combinator module can take multi-hot combinator or deep set combinator. The paper mainly applies their method to the biology data.

**Questions:**

Why do you choose Makkuva's dual formula instead of the primal formula? Do you want to get bi-directional maps?

Figure 1 is a bit confusing, especially 1b, according to the covariate condition experiment, when $c_i$ changes, $\mu_i$ also changes, but in fig1b, all $c_i$ corresponds to the same $\mu_i=\mu$.

Some questions about experimental results:
- In Figure 5, 6, you use one-hot embedding for unknown combinations, but I thought one-hot doesn't generalize to unknown action. Does "unknown combinations" mean combining existing perturbations instead of adding unseen perturbations? If that's the case, why not add an example for combinations of unknown perturbations?

- In Figure 6, a kind of conflicting result is that one-hot embedding is working better than moa embedding. Does this mean moa embedding is not necessary?

- Is the combinator module only used when you have the action of combination?

- The source distribution $\mu_i$ are the same for $\forall~ i$, which is just the control cells, except for the covariates condition?

Misc:

Figure 6 is out of margin border.

Typo: Line 217: where "M" is initialized ..

Line 131: Please write ||x||^2 in front of the backslash to avoid confusion.

The title of sections D.1, D.2 has a format issue: three dots.

Row 286: The figure pointers are not correct

In equation below line 555, sup shouldn't involve f*.

Caption of Fig 5, "one" instead of "ohe"

There exists one concurrent paper about the conditional transport map. It's up to the authors to add it to the literature part or not. "Asadulaev A, Korotin A, Egiazarian V, Burnaev E. Neural Optimal Transport with General Cost Functionals. arXiv preprint arXiv:2205.15403. 2022 May 30."

**Limitations:**

It asks the contexted measure to be paired. I also conjecture it's not easy to extend to imbalanced contexts, e.g. the two distributions have different number of labels.

**Strengths And Weaknesses:**

Strengths:

The paper parameterizes the dependence over the contexts in a flexible way. This makes their framework applicable to not only scalar labeled data but more complex ones, like multi-label (actions) or soft labels (covariates). Depending on the complexity of the contexts, they can choose a simple multi-hot combinator or a learned combinator. If using a learned combinator, they can generalize to unseen data. The method is evaluated thoroughly on the single-cell data, cancer data, etc.

Weak points:

Literature: 1) The technique of initializing ICNN to be an identity map has been introduced in [1], especially the quadratic layer technique in DenseICNN. 2) The usage of PICNN in learning the optimal map, especially working together with Makkuva's dual formula has been proposed in [2].

I'm worried about the writing. The fundamental loss function and algorithm are postponed to the supplementary material. To me, this is not proper. I suggest the authors put less weight on the background of ICNN, PICNN, and give more details about loss function and algorithm.

If I understand correctly, when you run ICNN OT, you just put data from all $\nu_i $ (or $\mu_i$) into a combined distribution $\nu$ (or $\mu$), and learn the map between $\mu$ and $\nu$. If this is the case, when all $\mu_i$ are equal, the pushforward distribution by ICNN OT map is fixed, which doesn't depend on the context. To me, this is an unreasonable baseline when all $\mu_i$ are equal. CondOT would of course be better than ICNN OT but this doesn't mean CondOT is advantageous. I think a good baseline is to calculate ICNN map for each $(\mu_i , \nu_i)$ pair. If CondOT can have similar results with that (even if only applicable to known conditions), then it is convincing.

[1] Korotin A, Egiazarian V, Asadulaev A, Safin A, Burnaev E. Wasserstein-2 generative networks. arXiv preprint arXiv:1909.13082. 2019 Sep 28.

[2] Fan J, Taghvaei A, Chen Y. Scalable computations of Wasserstein barycenter via input convex neural networks. arXiv preprint arXiv:2007.04462. 2020 Jul 8.

---

> ### Author Response · Authors · 2022-08-02
> **Reply to Reviewer kDhV (1/3)**
>
> Thank you for your detailed feedback and your insightful remarks!
>
> > **I think a good baseline is to calculate ICNN map for each $(\mu_i, \nu_i)$ pair.**
>
> Thanks a lot for raising this point. We have added an additional baseline in which we compare CondOT against an ICNN OT model trained on each context individually (i.e., distinct models for each context for each $(\mu_i, \nu_i)$ pair). The additional experiment and added baseline are described in more detail in the post **Response to All Reviewers** and demonstrate that CondOT predicts perturbation responses as well as a method purely trained on individual conditions, while still being able to generalize (see Table 1 in the paper).
>
> Training distinct models for each context for each $(\mu_i, \nu_i)$ pair is of course only possible if all conditions are distinctly known and we do not aim to generalize to unknown contexts. We are currently working on a problem with clinicians in which the provided context information is more complex and not separable into distinct groups on which individual ICNN OT models could be trained.

---

> > ### Author Response · Authors · 2022-08-02
> > **Reply to Reviewer kDhV (2/3)**
> >
> > You mention 3 limitations of our work that concern mostly credit assignment. Thank you for these references.
> >
> > > **1.  The technique of initializing ICNN to be an identity map has been introduced in [1], especially the quadratic layer technique in DenseICNN.**
> >
> > Many thanks for this reference, you are completely right. We found out about the identity initialization proposed in the Appendix of [1] shortly after the submission.  We have corrected our working version and now reference [1] prominently in that part. As you stated, [1] also proposes a quadratic (Dense)  layer, which the authors turn into a low-rank matrix given the high-dimensional image applications they consider. When refreshing our review, we also noticed that introducing a quadratic layer was also hinted at in [4, Appendix C.1].
> >
> > Compared to [1], our contribution and motivation to introduce these initialization schemes were:
> >
> > * To initialize that quadratic layer with a psd matrix AA’ that recovers an affine map informed by Gaussians. To the best of our knowledge, we believe this is completely new and very practical. While identity initialization can prove very useful for generic objectives for convex functions (e.g., training JKO), the Gaussian initialization works better to compute OT duals.
> > * To go one step further, see beyond ICNNs and encode these initializations within PICNNs (Fig. 3), handling multiple Gaussian approximations.
> >
> > Finally, there are a few important differences between our initialization and that of [1, Appendix B.1 and C.1]: Because [1] is set in the context of images, initialization schemes in [1] build on pretraining. Our motivation being single-cell data, we can use identity or Gaussian approximations directly without any additional steps (see Fig. 8 in Appendix C.1 of the updated paper for visualization and the post **Response to All Reviewers** for more details on the additional comparison between the initialization schemes).
> >
> > > **2.  The usage of PICNN in learning the optimal map … has been proposed in [2].**
> >
> > Thanks for this reference that we were unaware of, we will add it to our draft. We have given credit to the authors of [2] for experimenting with PICNNs in the context of OT estimation.
> >
> > A few facts should nuance claims that our paper is less novel as a result:
> >
> > *  PICNNs appear in Appendix B of [2] and not in the main paper, where their main proposal uses exclusively ICNNs.
> > * We gather from their experimental results, and their choice to move that part to the Appendix, that their PICNN did not work well enough, whereas the ICNNs worked all right for this task.
> > * This impression is reinforced by the fact that in their PICNN experiments, their context variable is a vector in the simplex (dimension 2), and they compute the barycenters of 3 Gaussians in d=2.
> >
> > One should contrast these facts with the setup in our paper: ICNNs do not work well for our purpose, and we use experiments in high dimensions on several contexts and leverage PICNNs to beat other baselines.
> >
> > > **3. There exists one concurrent paper Asadulaev, Korotin, et al. (2022) [3] about the conditional transport map.**
> >
> > We are happy to add any reference to the concurrent paper in our draft. For obvious reasons, we believe this should not be listed in the “limitations” section of your review, and respectfully ask you to move to another section of your review. There are a few crucial differences with Asadulaev, Korotin, et al. (2022) [3] that are worth clarifying. [3] proposes a framework to compute OT maps for general cost functionals, a setting also proposed by Fan et al. (2021) [14], and can be applied to learning maps that preserve the class-wise structure of data. This indeed partially resembles our covariate setting and presents an interesting alternative perspective on the problem. CondOT, however, is a general framework for learning the family of convex potentials by parameterizing a hypernetwork, which for a context $c$ returns the corresponding Monge map. This framework operates on contexts $c$ of various nature, not only when having a distinct class / covariate structure on the data, i.e., different actions simply "act different" on the entire source input measure $\mu$.

---

> > > ### Author Response · Authors · 2022-08-02
> > > **Reply to Reviewer kDhV (3/3)**
> > >
> > > > **Responses to Questions and Remarks**
> > >
> > > * **Why do you choose Makkuva's dual formula instead of the primal formula?**
> > >
> > >     Both use-cases are possible but from previous experience learning the mapping, the dual formulation shows slightly better stability during training, notably with the initialization we consider (for which both directions can be initialized using Gaussian approximations).
> > >
> > > * **You use one-hot embedding for unknown combinations, but I thought one-hot doesn't generalize to unknown action.**
> > >
> > >     When using the one-hot embedding module, we can aim for predicting **unseen** **combinations** of perturbations that were **previously seen in isolation**.
> > >
> > > * **It’s a conflicting result is that `one-hot` embedding is working better than the `moa` embedding.**
> > >
> > >     No, the one-hot embedding setting is a much simpler case than a mode-of-action embedding. In contrast to the one-hot embedding, the mode-of-action setting can also generalize to unseen perturbations (one-hot can only handle previously unseen combinations).
> > >
> > > * **Is the combinator module only used when you have the action of combination?**
> > >
> > >     Yes, we apply the combinator only if necessary. So the combinator is used in Fig. 5b and c, Figure 6 b and c, Figure 7, and Figure 11.
> > >
> > > * **Is the source distribution $\mu_i$ the same for all $i$, which is just the control cells, except for the covariates condition?**
> > >
> > >     In our experiments, when conditioning on scalars and actions we only assume a single source distribution. In the setting where the context is a covariate, we indeed assume different source distributions $\mu_i$ for different contexts $c_i$. As outlined above, however, you might encounter a different number of source or target measures (see Fig. 1 in the updated PDF).
> > >
> > >
> > > Thanks a lot for pointing out spelling and formatting errors. We have addressed them in the updated PDF.

---

> > > > ### Comment · Reviewer_kDhV · 2022-08-06
> > > > **Reply to authors**
> > > >
> > > > I appreciate and esteem the detailed replies. Just one number can be further checked: In Table R2 dosage 10, the loss of ICNN OT (on selected condition) is larger than CondOT.
> > > >
> > > > I've also increased my score and modified my review of the literature.

---

> > > > > ### Author Response · Authors · 2022-08-06
> > > > > **Follow-Up to "Reply to Authors"**
> > > > >
> > > > > We thank the reviewer for taking the time to read our rebuttal, for their appreciation of our work, and for updating their score.
> > > > >
> > > > > Regarding the better performance of CondOT for dosage 10: This is indeed a good point, and we might have been a bit misleading when calling the performance achieved by ICNN’s trained individually on each context a *lower bound*.
> > > > >
> > > > > In fact, we *do* expect CondOT to perform better than ICNNs trained independently in small data regimes. This is the *multi-task* aspect we plan to stress more in the final version. By *multitask* we refer to the usual definition:
> > > > >
> > > > > > *Multitask Learning is an approach to inductive transfer that improves generalization by using the domain information contained in the training signals of related tasks as an inductive bias. It does this by learning tasks in parallel while using a shared representation; what is learned for each task can help other tasks be learned better* (see https://en.wikipedia.org/wiki/Multi-task_learning).
> > > > >
> > > > > CondOT is a good example of a multitask algorithm: We pool *samples of all contexts* to train a single PICNN with a single parameter $\theta$, whereas each ICNN we train for a specific context (here, as dosage), can only look at one subset of the data, which is typically much smaller than all samples pooled together; for the example which we proposed in the rebuttal we have:
> > > > >
> > > > > | **Dosage**                     | **10** | **100** | **1000** | **10000** | **All** |
> > > > > |----------------------------|--------|---------|----------|-----------|---------|
> > > > > | **\# Samples in Target** | 642    | 671     | 696      | 612       | 2621    |
> > > > >
> > > > > So taking things to the extreme, if the sample sizes $n$ of each context are small, CondOT should beat ICNN baselines trained individually just because of the larger sample size obtained thanks to the multitask principle.
> > > > >
> > > > > We hope that this improves understanding.

---

### Official Review · Reviewer_zoPg · 2022-07-07

**Rating:** 6
**Confidence:** 2
**Soundness:** 2 fair
**Presentation:** 3 good
**Contribution:** 2 fair

**Summary:**

This paper proposes to learn a single Monge map $T$ between different pairs of probability measures $\{(\mu_i,\nu_i)\}$ such that the $i$th particular transport depends on a context function $c_i$, which corresponds to $T(c_i)$#$\mu_i \approx\nu_i$. The Monge map is parameterized by a convex neural network with partial input.


**Questions:**

- A bottleneck in your method seems to be the initialization. Could you elaborate more on why the proposed initializations (identity or Gaussian) work? Are there any theoretical results on this?

- You could specify that the measures you are considering are point clouds.

**Limitations:**

The underlying assumptions needed on the pair of probability measures for the method to work (in particular the initialization) are missing in my opinion.


**Strengths And Weaknesses:**

I find the idea of the paper, which consists of generalizing the learning of a transport map to several pairs of measures, which are first transformed by a "context" function, well justified in terms of application.

However, as far as I understand the method, several issues and limitations seem to arise, which I develop in the following points.
- An important part of the paper is to propose an efficient initialization. It is not clear to me why the initialization with Gaussian potential would work well in practice, for general dataset of measures that cannot be approximated by Gaussian distributions. It seems that there is a hidden assumption about the "Gaussian" geometry of the probability measures.
- Since by the properties of the pushforward operator one has for a deterministic function $c_i:\mathbb{R}^d\to\mathbb{R}^d$ that $T_\theta(c_i)$#$\mu_i$ is equivalent to $T_\theta$#$(c_i$#$\mu_i)$, a framework of your proposal could correspond to finding a unique application (or Monge map) that maps (approximately) each $c_i$#$\mu_i$ to $\nu_i$. Therefore, for this to be possible, I would say that the pairs $(c_i$#$\mu_i,\nu_i)$ must share a strong structure or relationship for all $i$. This is the case in the examples shown in Figure 1.

As a final comment, the proposed paper is very applied and the contribution is limited in my opinion, since the use of a convex neural network to parametrize a transport map has been widely developed in the literature.

Minor comments:
- l.99 : problem in the syntax of the sentence.
- l.128 : "Figure 3" should be "Figure 1", same l.144.
- Fig. 7 is not referenced in the text.
- Would a natural continuation of this work would be to consider normalising convex flow for the invertible Monge map in order to have similar uncertainty for both $\mu_i$ and $\nu_i$? This could be done based on the paper [Coeurdoux et al., 2022], which uses normalising flows to approximate a Monge map between a pair of empirical distributions, and then generalizes it to other pairs of point clouds

---

> ### Author Response · Authors · 2022-08-02
> **Reply to Reviewer zoPg (1/2)**
>
> Thanks for your review. We think you have misunderstood our work, since your first sentence reads:
>
> > **I find the idea of the paper, which consists of generalizing the learning of a transport map to several pairs of measures, which are first transformed by a "context" function, well justified in terms of application.**
>
> This is not the idea of our paper. In our work, context variables $c_i$ are **not** functions, they are more simply labels in a set (l. 24, l. 51). We provide a detailed discussion in l. 98-105 of what such labels can be (real numbers in [0, 1], vectors, discrete structures encoded as embeddings, etc.). At no point in the paper do we assume that contexts interact directly with the measures. Instead, they modulate maps.
>
> > **An important part of the paper is to propose an efficient initialization. It is not clear to me why the initialization with Gaussian potential would work well in practice, for a general dataset of measures that cannot be approximated by Gaussian distributions.**
>
> We insist that in the context of neural OT, initializing with Gaussian quantities simply means initializing our ICNN / PICNNs so that they are affine. It’s reasonable to think those maps should mimic, at time 0, something that gets the mean and covariances of the data right, and that is our contribution.
>
> The obvious alternative, using random weights initializations, is problematic: it is compounded by the fact that gradients (w.r.t $x$) of randomly initialized NN are extremely noisy and unstable. The bar is therefore not to have good performance directly, but to have a reasonable starting point, and using Gaussianity is just step 0.
>
> We added an additional experiment to the Appendix to exactly demonstrate this. Appendix C.1 of the updated paper displays the initializations at iteration 0 as well as the improved training convergence and stability throughout the optimization. We hope these examples help form a better intuition. For more details, see the post **Response to All Reviewers**.
>
> > **Since by the properties of the push forward operator one has for a deterministic function** $c_i: R^d \rightarrow R^d$ **that** $T_\theta(c_i)\sharp\mu_i$ **is equivalent to** $T_{\theta}\sharp(c_i \sharp \mu_{i})$ , **a framework of your proposal could correspond to finding a unique application (or Monge map) that maps (approximately) each** $c_i \sharp \mu_i$ to $\nu_i$**. Therefore, for this to be possible, I would say that the pairs** $(c_i\sharp\mu_i,ν_i)$ **must share a strong structure/relationship for all $i$.**
>
> We feel there might be a misunderstanding and you might have conceived our work on a first reading as an improved neural OT estimator that uses compositions of maps. As mentioned earlier, contexts are _not_ functions that we use to push forward $\mu$. A way to clarify this misunderstanding is to think of our work as a _hypernetwork_ (l. 53, 137): We are learning neural networks that output an OT neural network whose weights are entirely conditioned on $c$, as mentioned in the paragraph below line 106.
>
> While we have thought about parameterizing the problem the way you define it, as a composition, this approach would not generalize to new contexts, as we explain in (l. 17-27). For instance, given two treatments A, and B, and experiments showing the impact of A and B, the approach you suggest would allow modeling independently context map $c_A$ and $c_B$ in addition to the $T_\theta$ you mention, but would not allow predicting the effect of treatment {A+B} in a principled way (since we cannot estimate generalize a $c_{A,B}$ from there).
>
> > **The contribution is limited in my opinion, since the use of a convex neural network to parametrize a transport map has been widely developed in the literature.**
>
> We respectfully disagree with your assessment. We do not use a convex neural network to parameterize a transport map, we use a partially convex neural network, we believe this is an important distinction (l. 107 and above). This misunderstanding is also expressed in your summary, i.e., when you state “Monge map is parameterized by a convex neural network with partial input”. The neural network has two distinct inputs. The PICNN is thereby convex w.r.t. to its first input, but not convex w.r.t. its other input, hence partially input convex (l.110~125). There are, to our knowledge, no applications of this idea in the literature, apart from the preliminary attempt given in Appendix B of [2], as pointed out by Reviewer kDhv.
>
> > **A continuation of this work would be to consider normalizing convex flow for the invertible Monge map in order to have similar uncertainty for both $\nu_i$ and $\mu_i$ … based on the paper by Coeurdoux et al. (2022) [12].**
>
> Thanks for this suggestion. We will certainly add this recent contribution to our bibliographic review. We believe there are, for reasons detailed in the paper and in our answers here, several differences, and that their approach cannot be used here.

---

> > ### Author Response · Authors · 2022-08-02
> > **Reply to Reviewer zoPg (2/2)**
> >
> > > **Responses to Questions**
> >
> >
> >
> > * **A bottleneck in your method seems to be the initialization. Could you elaborate more on why the proposed initializations (identity or Gaussian) work? Are there any theoretical results on this?**
> >
> >     Thanks for the question. We believe this is quite the opposite. The initialization helps to train ICNN / PICNN, which are known to have a harder time estimating the transport in higher dimensions.
> >     As mentioned above, we have added a synthetic experiment to the paper (see Appendix C.1 of the updated PDF and the post **Response to All Reviewers**).
> >
> >     Regarding theoretical results, the Gaussian approximation-based initialization scheme is based on the closed-form solution of OT between Gaussians. More details on this can be found in Peyré and Cuturi (2019) [13, Remark 2.31].
> >
> >
> > * **The underlying assumptions needed on the pair of probability measures for the method to work (in particular the initialization) are missing**.
> >
> >
> >     We use real data for single-cell genomics experiments. We hope we have convinced you that the Gaussian initialization is just used to avoid the pitfalls of noisy initializations that result in very noisy gradients, as shown in the experiment.
> >
> > Thanks a lot for pointing out spelling and formatting errors. We have addressed them in the updated PDF.

---

> > > ### Comment · Reviewer_zoPg · 2022-08-08
> > > **Rebuttal**
> > >
> > > Thank you for your very much detailed answers and various modifications of the paper. In particular, the clarifications on initilization are really appreciated.
> > >
> > > I have read the other comments and responses, the clarifications on your formulation in terms of hypernetwork is convincing and the generalization of your method to unseen context values is clearer. The link made with multi-task learning is also interesting.
> > >
> > > As I stated in my confidence score, this paper is quite outside my area of expertise, and your responses were welcome. Therefore I am willing to increase my score accordingly.

---

> ### Author Response · Authors · 2022-08-07
> **Follow-Up**
>
> Dear Reviewer,
>
> We understand the time window to interact with reviewers is very short this year, and only lasts up to next Tuesday. Could you please confirm that you have had a chance to look at the new elements we have provided in the rebuttal?
>
> We do hope that the rebuttal has resolved all of the concerns raised in your review. In particular, we tried to clarify a possible misunderstanding of the role played by context variables $c$ in our architecture. Our goal, as explained below and in the original paper, is not to use contexts to add a layer sandwiched between the ICNN and the input $x$ as you mentioned, but truly have $c$ be intertwined with the parameters of the ICNN to form a **P**ICNN. We have also addressed the fact that Gaussianity of data is not needed for our approach to work, but rather a reasonable initialization for (P)ICNNs.
>
> Thank you very much in advance for kindly taking the time to look at our arguments below. We understand this may not be the most convenient period of the year to carry out this dialogue (especially if you are in the northern hemisphere), but we look very much forward to hearing your reaction to the responses we have provided below.
>
> Thank you again for your time.

---

### Official Review · Reviewer_P7bD · 2022-07-08

**Rating:** 6
**Confidence:** 3
**Soundness:** 3 good
**Presentation:** 3 good
**Contribution:** 2 fair

**Summary:**

This work concerns the problem of learning the optimal transport (Monge) map that push-forwards between a given pair of measures coupled with a given context. They propose a framework that composes of an input convex neural network and a context embedding, as well as two ways to effectively initialize the network. Empirically, they show that the proposed model can predict and generalize well in the application of population dynamics prediction coupled across various types of contexts.

**Questions:**

- In the experiments, the comparison with ICNN OT seems to be unfair as the whole context information is discarded. What happens if you feed the embedded context directly to the ICNN (not PICNN) or maybe try some trivial ways to incorporate the context (e.g., building a network that outputs $h(f(x), g(z))$ where $f$ is an ICNN and $g$ is a neural net and $h$ is some linear combination) - will it still perform poorly?

- Should there be some out-sample results in the setting of covariate conditioning in the experiments?
- Why are the UMAP embeddings of data in Figure 7 different in three plots? Should they be the same given the same data?
- Figure 7 appears in the main text without being referenced.

**Limitations:**

The discussion on limitations is inadequate to some extent (see comments above). It seems to be no potential negative societal impact.

**Strengths And Weaknesses:**

- The paper is well-written and well-presented.
- The given answer to the question "how to make use of additional contexts in the traditional framework of ICNNs for learning OT Monge maps" is straightforward: feeding embedded contexts as an additional input to the neural network. The challenge appears to be the design of a network that is only convex in the main input (support of the source measure) and not in the context input (supposedly for more expressiveness - but this is not discussed in the paper). The proposed design (called PICCN) of such a network is novel.
- Another difficulty is said to lie in the training of such an architecture, for which the authors propose two straightforward approaches for good initialization, but there is no experiment showing the effectiveness of both methods compared to just initializing randomly. Moreover, there is also no discussion/recommendation on which initialization to choose in practice. From the experiments, it seems that the Gaussian initialization is slightly better than the identity one, but in my opinion, it is due to the former carrying more data information through the Gaussian approximation). In addition, I believe that there is likely not much difference between these two when the source distribution is complex).
- The generalization claim of the proposed method is not strongly supported in some experiments. For example, in Table 1, the in-sample errors and out-sample errors are consistent for all considered methods, so it cannot be said that CondOT generalizes better.
- There is a wide range of applications for the proposed method in modelling population dynamics given contexts, which is highly practical. Moreover, the results seem significant (as there is a substantial improvement over the SOTA).

---

> ### Author Response · Authors · 2022-08-02
> **Reply to Reviewer P7bD (1/2)**
>
> > **A difficulty is said to lie in the training of such an architecture, for which the authors propose two straightforward approaches for good initialization, but there is no experiment showing the effectiveness of both methods compared to just initializing randomly.**
>
> We have added a synthetic experiment to the paper to demonstrate the effect of both initialization methods on training convergence and stability (details in the **Response to All Reviewers** and in Appendix C.1 of the updated PDF). You mention “that there is likely not much difference between these two when the source distribution is complex”. We insist that in the context of neural OT, it’s reasonable to think parameterized maps should mimic, at time 0, something that gets the mean and covariances of the data right, and that is our contribution. When source and target measure are similar, one can see that our Gaussian initialization defaults back to an initializer that is very close to the identity presented in l. 221-230. And as demonstrated in Fig. 8b of Appendix C.1 in the updated PDF, both initialization schemes similarly improve training convergence and robustness and thus greatly optimize training.
>
> > **The generalization claim of the proposed method is not strongly supported in some experiments. For example, in Table 1, the in-sample errors and out-sample errors are consistent for all considered methods, so it cannot be said that CondOT generalizes better.**
>
> CondOT’s generalization ability for the settings displayed in Table 1 (conditioned on scalar and covariate) is indeed subtle as the chosen context potentially has less strong effects on the source distribution. When trained on all but one context, the performance of ICNN OT will likely not drop too drastically between the in-sample and out-of-sample settings. CondOT, however, is outperforming in both settings with consistently smaller variations than the baselines. Its generalization capabilities become more evident in settings where the context induces a much stronger condition, i.e., when the context is an action or combinations of actions, as settings displayed in Fig. 5, 6, and 7.
>
> > **The comparison with ICNN OT seems to be unfair as the whole context information is discarded.**
>
> This is indeed a good point. We have added an additional baseline in which we compare CondOT against an ICNN OT model trained on each context individually (i.e., individual models for each context). The additional experiment and added baseline are described in more detail in the post **Response to All Reviewers** and demonstrate that CondOT predicts perturbation responses as well as a method purely trained on individual conditions, while still being able to generalize (see Table 1 in the paper).
>
> Training individual models for each context is of course only possible if all conditions are distinctly known. We are currently working on a problem with clinicians in which the provided context information is more complex and not separable into distinct groups on which individual ICNN OT models could be trained.
>
>
> > **What happens if you feed the embedded context directly to the ICNN (not PICNN) or maybe try some trivial ways to incorporate the context (e.g., building a network that outputs h(f(x),g(z)) where f is an ICNN and g is a neural net and h is some linear combination) – will it still perform poorly?**
>
> You are right in that we expect more ideas to emerge on how to define interesting conditional architectures, essentially conditional convex approaches.
>
> If we understand correctly, your first suggestion is to concatenate $c$ to $x$ in the ICNN at every evaluation, and the other is essentially to have two row vectors $a$ and $b$ and have  $output = a’ f(x) + b’ g(z)$.
>
> We believe the first suggestion is overkill, in the sense that it requires convexity on the label itself, which is not needed. The second suggestion is simply an ICNN in disguise (for the purpose of OT maps) since the gradient w.r.t. $x$ stays the same (the linear part on $g(z)$ disappears).
>
> After thinking for quite some time on our own, we found that the PICNN, as proposed originally by Amos et al. (2017) [5], was already a very expressive architecture. It can be interpreted (Eq. 4) as an ICNN where the weight matrices are modulated **column-wise** by the context, keeping desirable properties. This clarifies our intuition of our hyper-map $\mathcal{T}_\theta$ being a _hypernetwork_ (l. 53).

---

> > ### Author Response · Authors · 2022-08-02
> > **Reply to Reviewer P7bD (2/2)**
> >
> > > **Responses to Questions**
> >
> >
> > 1. **Should there be some out-sample results in the setting of covariate conditioning in the experiments?**
> >
> >     Yes, that would be indeed great. Unfortunately, the current data situation does not facilitate this as the existing covariate setting (cell types) only consisted of three context labels. We are currently working with clinicians on an extension of this method containing much more covariates. This, however, goes beyond the scope of this paper.
> >
> >
> > 2. **Why are the UMAP embeddings of data in Figure 7 different in three plots?**
> >
> >     We compute the UMAP embedding individually for the true target as well as predicted target cells to better visualize the mismatch between predicted and true perturbed populations of each method. We can change that but it will not change the message of these plots.
> >
> >
> > Thanks a lot for pointing out spelling and formatting errors. We have addressed them in the updated PDF.

---

> > > ### Comment · Reviewer_P7bD · 2022-08-04
> > > **Post-Rebuttal**
> > >
> > > I thank the authors for the detailed responses that adequately addressed my concerns. I have updated my score accordingly.

---

> > > > ### Author Response · Authors · 2022-08-06
> > > > **Thanks for taking the time to read our rebuttal.**
> > > >
> > > > We are grateful for you taking the time to read our rebuttal and for updating your score. Please do not hesitate to ask more questions should you have any.

---

### Official Review · Reviewer_sa6w · 2022-07-09

**Rating:** 7
**Confidence:** 3
**Soundness:** 3 good
**Presentation:** 3 good
**Contribution:** 3 good

**Summary:**

The paper considers the problem of learning a map $\mathcal T : \mathcal C \times \mathbb R^d \to \mathbb R^d$ such that given observations $(c_i, (\mu_i, \nu_i)) \in \mathcal C \times \mathcal P(\mathbb R^d)^2$ it satisfies ${\mathcal{T}(c_i)}$\#$\mu_i \approx \nu_i$ for each $ i $. The primary motivation is the following: suppose we observe unpaired biological data before and after treatment that can be represented as point clouds $\mu_i,\nu_i$ respectively. The 'context' or metadata of the treatment (e.g. dosage, type of drug etc) is encapsulated by $c_i$. The key aspects of the authors' approach to the problem is as follows.
1) Use optimal transport as a judicious way of selecting the push forward map.
2) Parametrize $\mathcal T=\mathcal T_\theta$ by a partially input convex neural network (nonconvex in the 'context' variable), motivated by the fact that the optimal transport map is the gradient of the unique convex function that trasports the source to the target.
3) Transform the context variables to vectors in Euclidean space by a generic 'embedding module' which are combined by the generic 'combinator' module (that can take an arbitrary number of contexts as input which allows for multiple treatments, the most challenging/interesting application).
4) Train the entire model by end-to-end gradient descent on a dual formulation of the OT problem over the parameters of the input convex neural networks, the embedding module and the combinator module.

The main contributions of the paper are:
1) Describing a state of the art framework that allows for unseen/composite contexts and utilizes the 'continuity' of the context space.
2) Describing an initialization procedure for the partially input convex neural network (with the aim of making training easier) such that the initial map is either (i) the identity (ii) the optimal transport between the Gaussian approximations.

**Questions:**

### Questions/Comments:
1) Personally I would prefer that the main text include the algorithm (Algorithm 1) and the training objective (display (12-13)) for easier readibility. A short derivation of the training objective could also be added. Although under the space-constraints this may not be possible.
2) In Section 2) clarify the relevant regularity requirements for the OT map to exist.
3) What is _+ in display (4) and bottom of page 6?
4) Line 129: The message here is not completely clear. The McCann interpolation parametrizes a geodesic, but in practice we have no reason to expect the curve $(t, (\mu_t, \nu_t))_{t\in[0,1]}$ to form a geodesic (if, say, $t$ is the dosage)?
5) Why not always take train_freq_f=2? (as in why is it beneficial to train $g$ more than $f$)
6) What motivates perturbation signature metric? It is much cruder than Wasserstein distance or MMD - is it commonly used in the literature? The explanation in the paragraph starting on line 672 could be simplified to simply the $\ell_2$ distance between means of prediction and truth.




### Typos/minor suggestions:
- Line 3: repeats the word optimization
- Line 18: from of
- Display (2) why introduce f instead of just use \psi (also display (5))
- Figure 6b perturbation vs combination unclear
- Line 73: unique convex potential
- Line 91: …we encode this by …
- Paragraph under line 106: …leveraging the context values as much as possible …
- same paragraph: …propose to learn a unique …. Unique in what sense?
- Line 108: …not in the context value…
- Line 119: replace ‘condition’ by ‘context’?
- Line 123: …and the weight matrices…
- Line 155: allows vs encourages
- Line 320: …that is able to…
- Line 538: ...in that …
- Line 548: …learn our model…
- Line 552: definition of \tilde\Psi seems wrong, shouldn’t be a product
- Line 553: $\int |f| d\mu < \infty$
- Line 572: parentheses
- Line 747: multiple typos

**Limitations:**

The limitations and potential negative societal impact have been adequately addressed.

**Strengths And Weaknesses:**

Strengths:
1) The problem is well-motivated, interesting and important.
2) The paper is clearly written and easy to follow.
2) The empirical results suggest that the outlined framework is superior to prior work on common tasks.
3) The outlined framework allows to generalize to unseens contexts and combine them, which wasn't possible with prior work.

Weaknesses:
1) A lot of the components of the framework have appeared in related literature in some form (partially input convex neural networks specifically for training OT maps with the same dual objective, permutation invariant network for combinator, encoding of contexts etc.). The main methodological contribution seems to be that (i) they don't train separately for each context, instead parametrize wrt the context continuously and thus leverage the intuition that OT maps for similar contexts should be similar and (ii) the initialization of the PICNNs.

2) One of the main contributions (highlighted in the paragraph on line 49) is the initialization of the PICNN. However, from the text it is unclear whether the initialization improves test performance or training speed. In Table 1 the difference in test performance between identity and Gaussian initialization doesn't seem statistically significant, and later in Figure 9 only the Gaussian model is shown. Intuitively one might expect the Gaussian initialization to perform better than identity, which in turn should be better than a randomly initialized networks, commenting on this would make this aspect of the paper more convincing.

---

> ### Author Response · Authors · 2022-08-02
> **Reply to Reviewer sa6w**
>
> > **A lot of the components of the framework have appeared in related literature.**
>
> To our knowledge, this is the first paper focusing on a _hypernetwork_ setting for optimal transport. This is also the first paper proposing a “conditional OT” model.  While we do build on existing bricks and network architectures, we do stress (l. 112) that the way we use these bricks was not intended by the proposal of (Amos et al., 2017 [5]).
>
> > **One of the main contributions is the initialization of the PICNN. However, from the text, it is unclear whether the initialization improves test performance or training speed.**
>
> This is an excellent comment. We have indeed focused on downstream tasks in the paper. We are convinced these initializations are robust and sturdy, and obviously much better than a random init.
>
> We understand the reviewer might want more evidence. To illustrate this, we have added a synthetic experiment to the paper to demonstrate the effect of both initialization methods on training convergence and stability (details in the **Response to All Reviewers** and in Appendix C.1 of the updated PDF). Your intuition that “the Gaussian initialization [...] perform[s] better than identity, which in turn should be better than a randomly initialized network” is partially correct: Focusing more deeply on our application, we want to note that a perturbation might not have a strong effect on a cell population. In that case, the identity initialization method is at initialization closer to the final result. Both initialization schemes, however, similarly improve training convergence and robustness and thus greatly optimize training (see Figure 8b in Appendix C.1). We hope that this improves understanding.
>
> > **Response to Questions and Comments**
>
> 1. **The main text should include the algorithm (Algorithm 1) and the training objective for easier readability.**
>
>     We agree with this. The training objective is provided in the Appendix but we have moved this back to the main body in the next version. We will likely extend that part.
>
> 2. **Add a clarification of the relevant regularity requirements for the OT map to exist.**
>
>     We have swapped our previous reference to Theorem 1.17 of Santambrogio (2015) [11] to Theorem 1.22, which is more precise. Existence requires that $\mu$ and $\nu$ have finite $L_2$ norm, and that $\mu$ puts no mass on $(d-1)$ surfaces of class $\mathcal{C}_2$. This has been added.
>
> 3. **What is _+ in display (4) and bottom of page 6?**
>
>     This is the _positive part _(or RELU) operator. Those terms within the bracket (which depend on $W^{zu}_k$) need to be positive in order to enforce the network to be input convex w.r.t. $x$. Note we have corrected a typo, that operator is not needed for terms in $W^{xu}_k$.
>
> 4. **The McCann interpolation parametrizes a geodesic, but in practice, we have no reason to expect the curve $(t, (μ\mu_t, \nu_t)), t \in [0, 1]$ to form a geodesic (if, say, t is the dosage)?**
>
>     Indeed, sorry for not being clear here. We believe that McCann’s interpolation provides the simplest example for practitioners to think in terms of a “bare-bones” trivial conditional OT model, and we wanted to stress this early on in that section. We have clarified this section in our paper.
>
> 5. **Why not always take train_freq_f=2?**
>
>     We adapted some of the hyperparameter choices from Makkuva et al. (2020) [4].
>
>
> 6. **What motivates perturbation signature metric?**
>
>     This is a metric more commonly used in the computational biology community and usually applied to analyze perturbation responses and predictions, e.g., [https://satijalab.org/seurat/reference/calcperturbsig](https://satijalab.org/seurat/reference/calcperturbsig). We will improve the description.
>
>
> Thanks a lot for pointing out spelling and formatting errors. We have addressed them in the updated PDF.

---

> > ### Author Response · Authors · 2022-08-09
> > **Follow-Up**
> >
> > Dear Reviewer,
> >
> > We understand the time window to interact with reviewers is very short this year, and only lasts until **today**. Would it be possible for you to confirm that you have had a chance to look at the new elements we have provided in the rebuttal? Should you require further clarifications, please do not hesitate to ask more questions.
> >
> > We do hope that the rebuttal has resolved all of the concerns raised in your review. In particular, we added an additional experiment highlighting the mechanism and effectivity of the introduced initialization methods and elaborated on the differences in our approach to the related work.
> >
> > We understand this may not be the most convenient period of the year to carry out this dialogue, but we look very much forward to hearing your reaction to the responses we have provided below.
> >
> > Thank you very much for your service,
> >
> > the authors

---

> > > ### Comment · Reviewer_sa6w · 2022-08-10
> > > **Reply to authors**
> > >
> > > Dear authors,
> > > I apologize for the delay, I was under the wrong impression I had more time for my reply. Thank you for addressing my questions, the changes to the paper are welcome. Having read the other reviews and replies, I update my score to 7.

---

> > > > ### Author Response · Authors · 2022-08-10
> > > > **Thank you for your time**
> > > >
> > > > Dear Reviewer,
> > > > Thank you for your time and for having read our rebuttal. We are grateful for your score increase.

---

### Official Review · Reviewer_CtVg · 2022-07-11

**Rating:** 7
**Confidence:** 4
**Soundness:** 3 good
**Presentation:** 4 excellent
**Contribution:** 2 fair

**Summary:**

The paper proposes a generalization of Monge maps to the case where multiple pairs of labeled measures $\\{(c_i,(\mu_i,\nu_i))\\}$ are to be matched by a single, global map. The desired map has the context label $c_i$ as an input, and for each $c_i$ it serves as the OT map from $\mu_i$ to $\nu_i$. The paper starts by setting preliminaries and describing the objective, then provides a detailed account of implementation using partially input convex neural nets, and concludes with several experiments on treatment effects of drug combinations.

**Questions:**

Major questions/concerns appear under `Weaknesses' above. Some smaller issues and specific questions are listed next:

1. **Eq. (2):** Should both $f^\star_{\mu,\nu}$ and $\psi^\star_{\mu,\nu}$ appear there? The authors seem to interchange between $f$ and $\psi$ for OT potentials throughout the paper. I suggest picking one symbol and sticking with it.

2. **Section 3.2, Item 1:** In the last sentence, $\psi_\theta$ is referred to as an 'OT map' but it should be 'OT potential'.

3. **Section 4, 1st paragraph:** The authors say 'it is able to approximately map a measure $\mu$...'. In what sense is the approximation here? Can this be quantified?

4. **Eq. (5):** Should write $f^\star_{\mathcal{N}_1,\mathcal{N}_2}(x)$ on the left-hand side, i.e., add the function argument to the notation.

5. **Fig. 6:** Why is there no comparison to ICNN OT [Makkuva et al., 2020] for that experiment?

6. **General suggestion:** Having the figures appear on the same page where they are discussed (or at most one page before) would improve presentation. Currently, figures are spread out widely across the text which makes reading harder. For instance, Fig. 4 appears on page 6 but is discussed the first time on p. 8. Jumping back and forth is quite inconvenient. I suggest clustering figures together, if necessary, and bringing them closer to the relevant text.

**Limitations:**

Coupling the results of the paper with some theory, as suggested above, would provide a principled account of limitations. Currently, the method seems to work well but there are no assurances on how far the framework can be pushed or when it might fail. The authors are clear about what is being proposed and are honest in their presentation of the results, but the limitations of the framework remain an item for further exploration.

I have no concerns about potential negative societal impact.

**Strengths And Weaknesses:**

**Strengths:**

* The paper is very well-written: clear, streamlined, and concise. The discussions and descriptions of ideas are detailed enough for the reader to get all the information they need, but at the same time they are not bolstered or too long. The diagrams are helpful as well and make for nice visual aids. The amount of background provided was also just right, making the paper self-contained and with a good flow. The technical writing and notation, although not a dominant part of this paper, are also to a high level. Overall, I enjoyed reading the paper.

* Another strength is that the problem is well-motivated. The authors successfully conveyed the importance of the proposed setting to the `treatment effect of combinations of drugs' problem and I am confident that their approach could come handy for additional applications.

* The approach is novel and the detailed implementation description serve as a nice contribution. The empirical results are also very encouraging and demonstrate how the conditional Monge map outperforms competing methods on several tasks. Altogether this makes for good practical/implementational contribution in the space of applied OT for inference and learning.

**Weaknesses:**

* I think that the paper can be improved by providing more theoretical context and justification for the conditional Monge map. Specifically, is there a natural (conditional) OT problem that one can set up for which the conditional Monge map emerges as a solution? Providing such a formulation would make the proposed framework more principled and would enable further theoretical exploration thereof.

* Proceeding from the above point, given such a formulation, the authors would be able to pursue various theoretical results that would strengthen and further justify the proposed approach. A few immediate ones that come to mind include:
1) Statistical approximation results and empirical convergence rates: how well do empirical conditional Monge maps (i.e., maps learned from data) approximate the population map? At what rate? what is the dependence of this rate on the ambient dimension (I would expect a curse of dimensionality effect to occur)?
2) Approximation by neural network results: For the population measures, wow well can a NN-based architecture approximate the actual conditional map?
3) Formal optimization guarantees: Can gradient/subgradient based methods be shown to converge to a local optimum or a stationary point for the conditional OT problem, and under what assumptions? Can we obtain a Sinkhorn-like algorithm to an entropically regularized version of this problem?

* Like mentioned above, I think that the considered problem is interesting and the writing is excellent, but the level of novelty of the paper is a concern. The authors do well to combine existing ideas together (PICNN, Brenier's theorem, Gaussian approximations, etc.) to obtain an end-product that is greater than the sum of its parts. That said, the number of original ideas in the paper is rather limited. I believe that accounting for some of the above points would significantly strengthen the paper.

---

> ### Author Response · Authors · 2022-08-02
> **Reply to Reviewer CtVg (1/2)**
>
> > **Can be improved by providing more theoretical context and justification for the conditional Monge map.**
>
> Many thanks for this comment. At the moment, our approach is indeed constructive; your suggestion to abstract out theoretical blocks makes sense.
>
> We propose to reframe our conditional OT recovery problem more carefully as a proxy task for a regression problem: the inputs are contexts $c_i$, and the outputs would be OT maps $T_i^*$. This agrees with the _hypernetwork_ scenario which we had started to stress in v0 of our draft (l.53, 137).  Our setup has, yet,  another layer of difficulty:  these OT maps $T_i^*$ are not observed. We only observe measures $\mu_i$ and $\nu_i$, the latter being assumed to be equal to $T_i^*\sharp\mu_i$. Unlike typical regression problems, we are therefore unable to have a direct loss.
>
> To solve that regression, we propose the following regression function: given a context $c$, we output  a network, (above l.107, $\mathcal{T}_\theta(c) : = x \rightarrow \nabla_x \text{PICNN}(x,c)$. We also define a loss: Because ground truth maps $T_i^*$ are not provided, we rely instead on data measures $\mu_i$ and $\nu_i$ to define a penalized dual formulation of an OT loss as in Makkuva et al. (2020) [4].
>
> > **Providing such a formulation would make the proposed framework more principled and would enable further theoretical exploration thereof.**
>
> We completely agree with you, this would be an ideal outcome. We have already started rewriting in that direction. Our task also has links with multi-task learning that might be relevant.
>
> > **Further theoretical results on …**
>
> 1. **… statistical approximation results and empirical convergence rates. How well do empirical conditional Monge maps (i.e., maps learned from data) approximate the population map? At what rate?**
>
>     The main result in the area is Hütter et al. (2021) [6] but its pessimism does not match practice. This is why entropic OT has been recently explored as well, notably the entropic map (Rigollet et al., 2022) [10].
>
>     A tangential problem concerns the universality of ICNNs [5, Theorem 1 in 7] to approximate any Brenier potential, and therefore support their use as OT map estimators.
>
>     We believe that as is often the case in OT, practice is well ahead of theory. Any result in our paper would require first a leap of understanding for our ability to understand how ICNNs can recover such maps Neural OT as a whole. This is a fairly nascent field, as shown by the overview given by Korotin et al. (2021)  (“Do neural OT solvers work?” [8]). We believe it would be very ambitious to extend these results to PICNN, but our work calls for such “hyper-ICNN” studies.
>
>
> 2. **… approximation by neural network results. For the population measures, how well can a neural network-based architecture approximate the actual conditional map?**
>
>    Previous literature has explored the properties and approximation capabilities of convex neural network architectures. In particular, Chen et al. (2019) [7, Theorem 1] provide a theoretical analysis that any convex function over a convex domain can be approximated in sup norm by a convex neural network. Further, Proposition 2 in Amos et al. (2017) [5] states that a PICNN network with $k$ layers can represent any FICNN with $k$ layers and any pure feedforward network with k layers. From this, one can argue that a PICNN network can approximate the class of partially input convex functions, required to approximate conditional Monge maps. We think stating this is not necessarily a strong contribution per se, but we believe following your remark that this might be interesting to add.
>
>
> 3. **… formal optimization guarantees. Can gradient/subgradient-based methods be shown to converge to a local optimum or a stationary point for the conditional OT problem, and under what assumptions? Can we obtain a Sinkhorn-like algorithm for an entropically regularized version of this problem?**
>
>     The Sinkhorn objective can indeed enter the picture to estimate the problem (i.e., replace the Makkuva loss with a Sinkhorn loss). We are aware of several teams exploring that alternative approach for ICNNs.
>
>     However, integrating a context in the recently proposed families of entropic maps [10] presents an interesting research avenue, but this would require a different paper.

---

> > ### Author Response · Authors · 2022-08-02
> > **Reply to Reviewer CtVg (2/2)**
> >
> > > **The method seems to work well but there are no assurances on how far the framework can be pushed or when it might fail.**
> >
> > This is a very good comment, and we agree that a _lower bound_ would significantly improve the claims of our paper. We added this lower bound as an additional baseline and discuss it in more detail in the post **Response to All Reviewers**. This additional experiment demonstrates that CondOT predicts perturbation responses as well as a baseline that was purely trained on individual conditions, while still being able to generalize (see Table 1 in the paper). In addition, Bunne et al., (2022) [9, Figure 2b, e] compare the ICNN OT framework to the truly observed target cells, i.e., the situation where cellular variations are only due to measurement noise (evaluated w.r.t. the MMD metric). They demonstrate the success of ICNN OT in single-cell perturbation response modeling and show that it reduces the loss to a similar level as an intersample variation.
> >
> > Similarly, we second that an analysis of when the framework will fail is crucial. One of the hypotheses of applying optimal transport-based methods to predict the response to perturbations is that a population entity's features, in our case cellular features,  _incrementally_ change upon perturbation. Thus, we can match unperturbed and perturbed cell populations based on the minimal effort principle, i.e., finding the optimal map corresponding to the lowest transport cost. For particularly strong, complicated perturbations or when measurements of the cells are too far apart in time, cellular multiplex profiles might change too drastically, violating OT assumptions and making it challenging to reconstruct the alignments between unperturbed and perturbed populations using optimal transport. This concern was also partially addressed in Bunne et al., (2022) [9, Figure 4f]. This bottleneck, however, does also apply to other baselines [9, Figure S10].
> >
> > > **Responses to Questions**
> >
> > * **In Eq. (2), should both** $f_{\mu, \nu}^*$ and $\psi_{\mu, \nu}^*$ **appear there? The authors seem to interchange between $f$ and $\psi$ for OT potentials throughout the paper.**
> >
> >
> >     It is a common practice in the literature to use two different notations for the Brenier potential and the Kantorovich dual potential (see e.g., the book by Santambrogio (2015) [11]) although they do coincide in the setup we present (Eq. 2). We agree that notations should be simplified in this section.
> >
> > * **The authors say ”it is able to approximately map a measure $\mu$ …”. In what sense is the approximation here? Can this be quantified?**
> >
> >     By approximately, we mean that $\mu$ is mapped onto a measure that has the same first/second order moments as $\nu$. This was written also in l. 60 but we clarified this in the main text.
> >
> > * **Why is there no comparison to ICNN OT (Makkuva et al., 2020 [4]) for the experiment in Figure 6?**
> >
> >     We do not provide experiments using ICNN OT because ICNN OT is not able to extrapolate to unknown perturbations (i.e., new context vectors $c_\text{new}$). We can try to add ICNN OT to the left panel.
> >
> >
> > Thanks a lot for pointing out spelling and formatting errors. We have addressed them in the updated PDF. We will rearrange the figures to align them better with the text in the next version of the paper.

---

> > > ### Comment · Reviewer_CtVg · 2022-08-05
> > > **Answer after review**
> > >
> > > I thank the authors for their detailed answers and promised modifications. Can they provide further details on planned additions/edits for the camera-ready version? Specifically, concerning the:
> > > * **Theoretical formulation --** what changes are planned and in which section?
> > > * **Their response to the comment they labeled `Further theoretical results on...' --** would the mentioned references and discussion be included in the paper; where?
> > > * **Their response to the comment about assurances for and/or pitfalls of the framework --** Perhaps discuss these issues in the conclusion and future work section?
> > >
> > > I am open to increasing my score but would like to understand the above first.

---

> > > > ### Author Response · Authors · 2022-08-05
> > > > **Follow-Up to "Answer after Review"**
> > > >
> > > > We thank the reviewer for reading our rebuttal. We are currently drafting a modified version to incorporate the changes, as we feel it might be more convincing to provide these changes in pdf, rather than here. However, this might take a bit of time, and given the short duration of the rebuttal here is a short answer to the 3 points raised:
> > > >
> > > > > **Theoretical formulation, what's changed?**
> > > >
> > > > Following the reviewer's suggestion, the changes would take place at the beginning of Section 3. They would involve of course citing all the works mentioned above (on the hardness of estimating maps, PICNN universality, etc.), but also providing a better motivation for our estimation. This would complement very naturally the algorithm:
> > > >
> > > > Rather than start directly with a constructive approach, we would state the following. Although the dataset that is provided consists in contexts and pairs of measures $\{(c_i, (\mu_i,\nu_i)\}$, imagine the OT maps $T_i :=T^\star_{\mu_i,\nu_i}$ are provided. This would result in a dataset $(c_i, T_i)$. A standard regression approach would be to estimate, using a parameterized family $\mathcal{T}_{\theta}$ the parameter minimizing a fit error
> > > >
> > > > $$\min_{\theta} \sum_{i=1}^N \ell( \mathcal{T}_\theta(c_i), T_i),$$
> > > >
> > > > where $\ell$ would be a suitable loss (e.g., $\ell_2$ using measure $\mu_i$). Although this would make sense, this is just an abstract exercise, since we do not have access to the $T_i$. Instead, and in accordance with experiments, we only have access to measures $\mu_i, \nu_i$. What we know, though, is, by Brenier, that $T_i$ should be the gradient of a given convex potential $\psi_i$ that solves the dual problem (2). Hence, to exploit this identity and to find a way to estimate $\theta$ still, the problem we propose to solve instead id
> > > >
> > > > $$\max_{\theta} \sum_{i=1}^N  \mathcal{E}_{\mu_i,\nu_i}(\psi(\cdot, c_i)),$$
> > > >
> > > > where $\mathcal{E}_{\mu,\nu}(\phi) = \int \phi^* d\mu +\int \phi d\nu$ is the standard dual energy defined in Eq. 2. This is the objective we optimize, with the added modification that we use Makkuva's approach [4] to handle Legendre transforms.
> > > >
> > > > > **Further theoretical results on ...**
> > > >
> > > > We plan to integrate all three comments. The first comment (hardness) will be part of the discussion above and more generally to motivate experiments with various sizes $n$ for the number of points. Comments on the universality of ICNN are already quoted in the paper but will give more prominent room, in addition to those known on PICNNs. We will mention that we do not see immediately how to apply an entropic (Sinkhorn) map approach to our framework.
> > > >
> > > > > **Their response to the comment about assurances for and/or pitfalls of the framework ...**
> > > >
> > > > We believe the individual ICNN OT results provide part of that answer, and we will seek more results in that vein. We plan notably to explore sample size $n$ trade-offs: for *purely observed* contexts, we expect that a small sample size will favor the pooling effect achieved by the PICNN, while for much larger $n$ we expect ICNN OT to be at least as good (still, with the limitation that it cannot generalize to unseen contexts). Another limitation we seek to explore is network size. We have been relatively straightforward on that end and have only used a very reasonable range of parameters that seemed to work, fortunately, off-the-shelf. We would like to show, if possible, changes in performance depending on that part.

---

> > > > > ### Comment · Reviewer_CtVg · 2022-08-06
> > > > > **Answer**
> > > > >
> > > > > Thanks for the detailed description of planned edits. I've increased my score by 1.

---

> > > > > > ### Author Response · Authors · 2022-08-07
> > > > > > **Thanks for sharing your comments after rebuttal**
> > > > > >
> > > > > > Thank you for spending the time to read our rebuttal and reconsidering your original score. We greatly appreciate your feedback and will use it as described above to improve the draft.

---

### Author Response · Authors · 2022-08-02
**Response to All Reviewers (1/3)**

We thank all reviewers for their extensive feedback and time invested in suggesting important improvements for our work.

Two reviewers list concurrent works (**zoPg** cites arXiv:2207.01246, **kDhV** cites arXiv:2205.15403). While we are convinced it is worth adding a reference to these papers, these papers appear currently in the “questions” and “limitations” sections of their reviews, which is problematic. We respectfully ask reviewers to put these suggestions elsewhere in their reviews and understand that these requests are sensitive asks given their privileged position as reviewers.

Our main paper has several changes and we updated the PDF. Here are our main highlights:


## Additional Baseline

In our initial submission, we compared CondOT against current state-of-the-art methods (CPA by Lotfollahi et al., 2021) and previous neural optimal transport-based methods used on single-cell data (ICNN OT by  Makkuva et al., 2020). As suggested by several reviewers, we **added another experiment and additional baseline** to challenge the performance of CondOT even further, in which we trained ICNN OT individually for each distinct condition. This baseline can be seen as a **lower bound** on what accuracy CondOT can reach based on modeling perturbation responses by parameterizing Monge maps.

In particular, we train CondOT and both baselines (ICNN OT and CPA) to predict the dosage-dependent perturbation response to two different drugs, Trametinib and Givinostat. As before, CondOT and CPA are conditioned on all dosages, while an overall ICNN OT model is trained (i.e., _ICNN on all conditions_).

In the additional baseline (the lower bound, i.e., _ICNN on selected condition_) we compute different ICNN OT models for each different dosage. While this would fail to generalize to new contexts and it requires all contexts to be distinctly known, this is, in a way, the best we can expect to achieve.

We believe the setting in which we condition on scalars is a good start because in this 1D setting for $c$, the inability to generalize is less critical (as opposed to predicting previously unobserved combinations of drugs).

The results are displayed in the Tables below (**Table R1** for an analysis in PCA space and **Table R1** for results on 50 marker genes in data space with 1000 genes). The results clearly show that CondOT predicts perturbation responses as well as a baseline that was trained purely on individual conditions, while still being able to generalize (see Table 1 in the paper). As often mentioned in the multitask learning literature, sharing of parameters (the PICNN) and conditioning seems to improve by increasing the effective sample size of the problem.

**Table R1.** Wasserstein loss of different methods in PCA space for the drugs Givinostat and Trametinib, where we conduct the analysis for different dosages individually on the dataset by Srivatsan et al., (2020).

|Dosage     |      10         |      100        |      1000       |       10000       |
|-------------------------------|-----------------|-----------------|-----------------|------------------|
|CPA                            |69.83 $\pm$ 5.521|71.34 $\pm$ 2.84 |79.31 $\pm$ 6.36 |141.71 $\pm$ 94.74|
|ICNN OT (on all conditions)    |85.94 $\pm$ 24.74|85.64 $\pm$ 24.56|87.41 $\pm$ 26.01|107.43 $\pm$ 54.81|
|ICNN OT (on selected condition)|68.99 $\pm$ 0.69 |68.47 $\pm$ 0.61 |66.04 $\pm$ 1.05 |56.35  $\pm$ 11.59|
|CondOT (Identity)              |70.86 $\pm$ 3.18 |70.73 $\pm$ 3.61 |71.25 $\pm$ 3.89 |61.94  $\pm$ 7.43 |
|CondOT (Gaussian)              |70.46 $\pm$ 1.91 |70.02 $\pm$ 1.78 |68.01 $\pm$ 0.01 |60.48  $\pm$ 9.06 |

**Table R2.** Wasserstein loss of different methods on the top-50 marker genes for the drugs Givinostat and Trametinib, where we conduct the analysis for different dosages individually on the dataset by Srivatsan et al., (2020).

| Dosage                          | 10                | 100               | 1000              | 10000              |
| ------------------------------- | ----------------- | ----------------- | ----------------- | ------------------ |
| Model                           |                   |                   |                   |                    |
| CPA                             | 13.75 $\pm$ 1.41 | 13.75 $\pm$ 0.93 | 15.81 $\pm$ 2.16 | 55.12 $\pm$ 58.14 |
| ICNN OT (on all conditions)     | 12.37 $\pm$ 1.66 | 12.53 $\pm$ 2.40 | 13.36 $\pm$ 3.02 | 31.02 $\pm$ 28.12 |
| ICNN OT (on selected condition) | 10.98 $\pm$ 1.15 | 10.37 $\pm$ 0.23 | 10.58 $\pm$ 1.93 | 20.55 $\pm$ 14.16 |
| CondOT (Identity)               | 10.54 $\pm$ 0.37 | 10.58 $\pm$ 0.04 | 12.13 $\pm$ 2.43 | 20.97 $\pm$ 15.02 |
| CondOT (Gaussian)               | 10.56 $\pm$ 0.45 | 10.54 $\pm$ 0.16 | 12.12 $\pm$ 2.26 | 21.30 $\pm$ 15.29 |

---

> ### Author Response · Authors · 2022-08-02
> **Response to All Reviewers (2/3)**
>
> ## Extended Analysis and Visualization of Different Initialization Methods
>
> We conducted an **additional analysis and comparison between the vanilla initialization and those introduced in this work**. For the analysis, we chose a synthetic experiment that not only allows us to demonstrate improvements in training convergence and robustness, but also facilitates simple visualizations of the ICNN output at training iteration 0 (before training), highlighting the differences between the vanilla, identity, and Gaussian approximation-based initialization schemes. Details on this can be found in the updated PDF in Appendix C.1 and Figure 8.
>
> In particular, Figure 8b demonstrates that the **newly introduced initialization schemes not only improve training convergence but also robustness** and thus greatly improve and simplify training. This holds for both, the identity and the Gaussian approximation-based initialization.

---

> > ### Author Response · Authors · 2022-08-02
> > **Response to All Reviewers (3/3)**
> >
> > ## References
> >
> > [1] Korotin A, Egiazarian V, Asadulaev A, Safin A, Burnaev E. Wasserstein-2 Generative Networks. arXiv preprint arXiv:1909.13082. 2019.
> >
> > [2] Fan J, Taghvaei A, Chen Y. Scalable Computations of Wasserstein Barycenter via Input Convex Neural Networks. arXiv preprint arXiv:2007.04462. 2020.
> >
> > [3] Asadulaev A, Korotin A, Egiazarian V, Burnaev E. Neural Optimal Transport with General Cost Functionals. arXiv preprint arXiv:2205.15403. 2022.
> >
> > [4] Makkuva, Ashok, et al. "Optimal transport mapping via input convex neural networks." International Conference on Machine Learning. PMLR, 2020.
> >
> > [5] Amos, Brandon, Lei Xu, and J. Zico Kolter. "Input Convex Neural Networks." International Conference on Machine Learning. PMLR, 2017.
> >
> > [6] Hütter, Jan-Christian, and Philippe Rigollet. "Minimax estimation of smooth optimal transport maps." The Annals of Statistics 49.2 (2021): 1166-1194.
> >
> > [7] Chen, Yize, Shi, Yuanyuan and Zhang, Baosen. "Optimal Control Via Neural Networks: A Convex Approach." International Conference on Learning Representations (ICLR) (2019).
> >
> > [8] Korotin, Alexander, et al. "Do Neural Optimal Transport Solvers Work? A Continuous Wasserstein-2 Benchmark." Advances in Neural Information Processing Systems 34 (2021).
> >
> > [9] Bunne, C., Stark, S., Gut, G., del Castillo, J. S., ... & Rätsch, G. Learning Single-Cell Perturbation Responses using Neural Optimal Transport. https://doi.org/10.21203/rs.3.rs-1805107/v1. Nature Portfolio (2022).
> >
> > [10] Rigollet, Philippe, and Austin J. Stromme. "On the sample complexity of entropic optimal transport." arXiv preprint arXiv:2206.13472 (2022).
> >
> > [11] Santambrogio, Filippo. "Optimal Transport for Applied Mathematicians." Birkäuser, NY 55.58-63 (2015): 94.
> >
> > [12] Coeurdoux, Florentin, Nicolas Dobigeon, and Pierre Chainais. "Learning Optimal Transport Between two Empirical Distributions with
> > Normalizing Flows." arXiv preprint arXiv:2207.01246 (2022).
> >
> > [13] Peyré, Gabriel, and Marco Cuturi. "Computational Optimal Transport: With Applications to Data Science." Foundations and Trends® in Machine Learning 11.5-6 (2019): 355-607.
> >
> > [14] Fan, Jiaojiao, et al. “Scalable Computation of Monge Maps with General Costs." arXiv preprint arXiv:2106.03812 (2021).

---

### Meta-Review · Area_Chair_8Wma · 2022-08-25

**Recommendation:** Accept
**Confidence:** Certain

**Metareview:**

The paper considers the problem of learning a single Monge map between different pairs of probability measures such that the particular transport depends on a given context. The reviewers have found the paper well written and motivated. In addition the approach is novel and interesting. I therefore recommend acceptance.

**Award:**

No

---

### Decision · Program_Chairs · 2022-09-14

Accept